# Olig3 regulates early cerebellar development

Elijah D Lowenstein[1†], Aleksandra Rusanova[2,3†], Jonas Stelzer[2],
Marc Hernaiz-Llorens[1], Adrian E Schroer[2], Ekaterina Epifanova[2,3],
Francesca Bladt[1], Eser Göksu Isik[2], Sven Buchert[1], Shiqi Jia[1,4], Victor Tarabykin[2,3],
Luis R Hernandez-Miranda[1,2‡*]

[1]Max-Delbrück-Centrum in the Helmholtz Association, Berlin, Germany; [2]Institute
for Cell Biology and Neurobiology, Charité Universitätsmedizin Berlin, Berlin,
Germany; [3]Institute of Neuroscience, Lobachevsky University of Nizhny Novgorod,
Nizhny Novgorod, Russian Federation; [4]The First Affiliated Hospital of Jinan
University, Guangzhou province, Guangzhou, China

**Abstract** The mature cerebellum controls motor skill precision and participates in other
sophisticated brain functions that include learning, cognition, and speech. Different types of
GABAergic and glutamatergic cerebellar neurons originate in temporal order from two progenitor
niches, the ventricular zone and rhombic lip, which express the transcription factors Ptf1a and
Atoh1, respectively. However, the molecular machinery required to specify the distinct neuronal
types emanating from these progenitor zones is still unclear. Here, we uncover the transcription
factor Olig3 as a major determinant in generating the earliest neuronal derivatives emanating from
both progenitor zones in mice. In the rhombic lip, Olig3 regulates progenitor cell proliferation. In
the ventricular zone, Olig3 safeguards Purkinje cell specification by curtailing the expression of
Pax2, a transcription factor that suppresses the Purkinje cell differentiation program. Our work thus
defines Olig3 as a key factor in early cerebellar development.

**\*For correspondence:**
luis.hernandez-miranda@charite.
de

†These authors contributed
equally to this work

**Present address:** ‡Institute for
Cell Biology and Neurobiology,
Charité Universitätsmedizin
Berlin, Berlin, Germany

**Competing interests:** The
authors declare that no
competing interests exist.

**Reviewing editor:** Roy V Sillitoe,
Baylor College of Medicine,
United States

## Introduction

The cerebellum develops from the dorsal aspect of rhombomere 1, a region known as the cerebellar
anlage that in mice becomes apparent at embryonic (E) day 9.5 (*Butts et al., 2014*; *Chizhikov et al.,
2006*; *Millet et al., 1996*; *Morales and Hatten, 2006*; *Wingate and Hatten, 1999*). This region con-
tains two distinct germinal zones, the rhombic lip and the ventricular zone, that generate all gluta-
tergic and GABAergic cerebellar neurons, respectively (*Alder et al., 1996*; *Hallonet et al., 1990*;
*Wingate and Hatten, 1999*; *Zervas et al., 2004*). Development of glutamatergic and GABAergic
cerebellar neurons largely depends on the differential expression of two basic helix-loop-helix
(bHLH) transcription factors: Atonal homolog one transcription factor (Atoh1) and Pancreas-specific
transcription factor 1a (Ptf1a) (*Gazit et al., 2004*; *Hoshino et al., 2005*; *Machold and Fishell, 2005*;
*Millen et al., 2014*; *Wang et al., 2005*; *Yamada et al., 2014*).

In the rhombic lip, Atoh1 directs the generation of three neuronal derivatives: (i) deep cerebellar
nuclei (DCN) neurons (between E10.5 and E13.5), (ii) external granule layer (EGL) cells (between
E13.5 and birth), which are the precursors of the internal granule layer cells that develop during early
postnatal life, and (iii) unipolar brush cells (between E15.5 and the first days of postnatal life) (*Ben-
Arie et al., 1997*; *Englund et al., 2006*; *Fink, 2006*; *Gazit et al., 2004*; *Machold and Fishell, 2005*;
*Machold et al., 2011*; *Sekerková et al., 2004*; *Yamada et al., 2014*). In the ventricular zone, Ptf1a
instructs the generation of Purkinje cells (between E11.5-E13.5) and all inhibitory interneurons,
including Golgi, Stellate, and Basket cells (between E14.5 and birth). Inhibitory interneurons are
characterized by the expression of the homeodomain transcription factor Pax2 (*Hashimoto and*

*Mikoshiba, 2003*; *Hoshino et al., 2005*; *Leto et al., 2006*; *Maricich and Herrup, 1999*). Although the ablation of *Atoh1* and *Ptf1a* severely impairs glutamatergic and GABAergic cerebellar neuron development (*Ben-Arie et al., 1997*; *Hoshino et al., 2005*; *Jensen, 2004*; *Sellick et al., 2004*), less is known about the molecular machinery required for the temporal specification of the different neuronal derivatives emerging from the two cerebellar neurogenic niches.

bHLH transcription factors are master regulators of progenitor cell differentiation during development and are critical players in neuron subtype specification in the nervous system (*Atchley and Fitch, 1997*; *Baker and Brown, 2018*; *Ben-Arie et al., 2000*; *Bertrand et al., 2002*; *Dennis et al., 2019*; *Dokucu et al., 1996*; *Imayoshi and Kageyama, 2014*; *Jones, 2004*; *Mattar et al., 2008*; *Ross et al., 2003*; *Sommer et al., 1996*). Among these factors, Oligodendrocyte factor 3 (Olig3) has been implicated in the specification of dorsally emerging neuron types in the hindbrain and spinal cord (*Hernandez-Miranda et al., 2017b*; *Liu et al., 2008*; *Müller et al., 2005*; *Storm et al., 2009*; *Zechner et al., 2007*). However, its molecular mechanisms and functions outside these regions have been less studied (*Shiraishi et al., 2017*; *Vue et al., 2007*). Although *Olig3* expression was previously reported during cerebellar development, its function there has not yet been explored (*Liu et al., 2010*; *Takebayashi et al., 2002*).

In this study, we sought to identify bHLH factors that contribute to the development of distinct cerebellar neuron types. We report here that Olig3 is crucial for generating the earliest rhombic lip and ventricular zone neuronal derivatives. Our lineage-tracing studies illustrate that the majority of DCN neurons, EGL, granule cells, as well as Purkinje cells emerge from Olig3+ progenitor cells. In contrast, few inhibitory interneurons had a history of *Olig3* expression. Ablation of *Olig3* results in severe cerebellar hypoplasia. In particular, we show that in *Olig3* mutant mice, most DCN neurons as well as half of the EGL cells, granule cells and Purkinje cells are not formed. In contrast, supernumerary inhibitory interneurons develop in *Olig3* mutant animals. Mechanistically, we show that Olig3 cell-autonomously suppresses the development of inhibitory interneurons in the ventricular zone. Olig3 is first expressed in ventricular zone progenitor cells and transiently retained in newborn Purkinje cells to curtail the expression of Pax2, a gene that we found to suppress the Purkinje cell differentiation program. We also show that Olig3 and its close family member Olig2 specify complementary Purkinje cell populations. Altogether, our data provide new insights into the molecular machinery that secures the correct development of cerebellar neurons.

## Results

### Olig3 is expressed in rhombic lip and ventricular zone progenitor cells during early cerebellar development

About 130 bHLH transcription factors have been found in humans and 117 in mice (*Skinner et al., 2010*; *Stevens et al., 2008*). To identify candidate bHLH factors that contribute to the generation of early versus late derivatives from the rhombic lip and ventricular zone, we first analyzed the expression patterns of 110 bHLH transcription factors annotated in the Human Genome Organization (HuGO; https://www.genenames.org/data/genegroup/#!/group/420) throughout mouse cerebellar development using publically available data from the Allen Developing Mouse Brain Atlas (https://developingmouse.brain-map.org). We found that 51 bHLH genes were expressed during cerebellar development, of which 27 were seen in progenitor niches (rhombic lip, ventricular zone, and/or EGL), and the remainder in postmitotic regions (*Figure 1A*; *Table 1* and *Figure 1—figure supplement 1*). In particular, 9/27 genes displayed differential spatial-temporal expression patterns in the rhombic lip (Atoh1 and Olig3 between E11.5-E13.5), EGL (Atoh1 and Neurod1 between E13.5-birth), and ventricular zone (Ptf1a, Ascl1, Olig3, and Olig2 between E11.5-E13.5; Ptf1a, Neurod6, Neurog1 and Neurog2 between E13.5-E18.5). The remaining (18/27) factors appeared to belong to either a common set of transcription factors expressed in all progenitor niches or they maintained their expression in a particular niche throughout cerebellar development (*Table 1*). Of particular interest was the expression pattern of Olig3, which has not been previously reported to have a function in cerebellar development. However, it is known to participate in the specification of defined hindbrain and spinal cord neurons (*Hernandez-Miranda et al., 2017a*; *Liu et al., 2008*; *Müller et al., 2005*; *Storm et al., 2009*; *Zechner et al., 2007*).

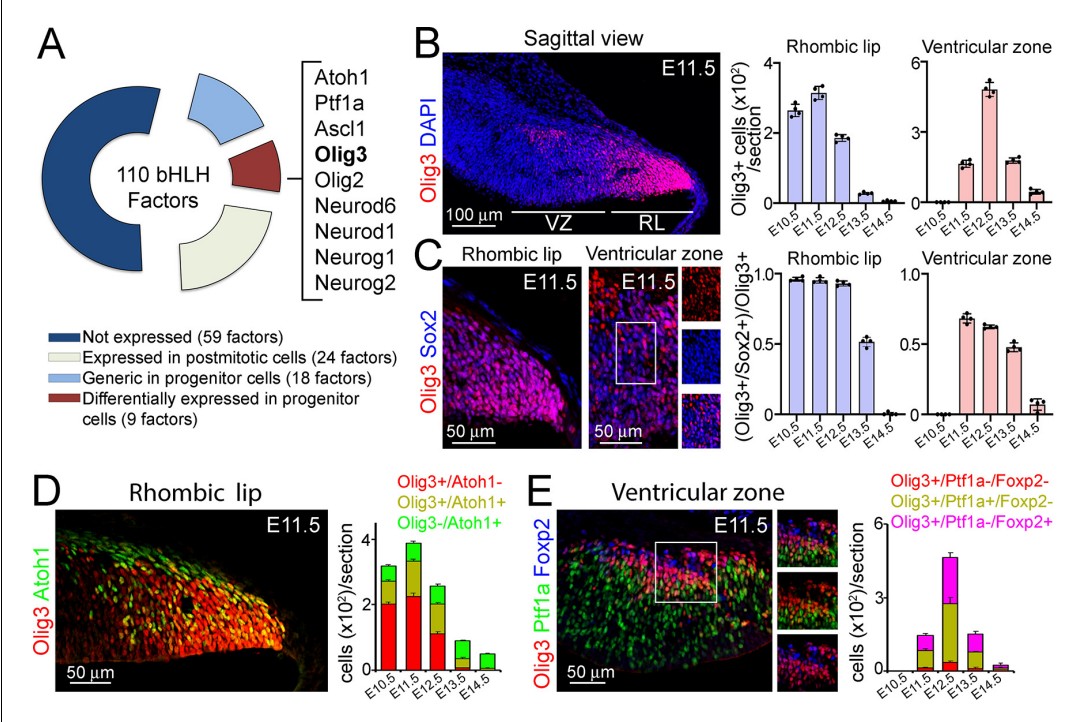

**Figure 1.** Olig3 marks rhombic lip and ventricular zone progenitor cells during early cerebellar development. (**A**) Doughnut chart illustrating the expression of 110 bHLH transcription factors during cerebellar development. For details on individual gene names within each category please see *Table 1*. (**B**) Left, a sagittal section of the cerebellum stained against Olig3 (red) and counterstained with DAPI (blue) at E11.5. Right, quantification of Olig3+ cells in the rhombic lip (RL) and ventricular zone (VZ) between E10.5 and E14.5. See also *Figure 1—figure supplement 2A*. The rhombic lip and ventricular zone domains were defined in this study according to the expression of Atoh1 and Ptf1a, respectively (see *Figure 1—figure supplement 2B*). (**C**) Left, magnifications of the rhombic lip and ventricular zone of a sagittal cerebellar section stained against Olig3 (red) and Sox2 (blue) at E11.5. The boxed area displayed in the ventricular zone is illustrated to the right of the main photograph. Right, quantification of the proportion of Olig3+ cells co-expressing Sox2 between E10.5 and E14.5. Dots in the graphs represent the mean of individual analyzed animals. (**D**) Left, immunofluorescence characterization of rhombic lip progenitor cells stained against Olig3 (red) and Atoh1 (green) at E11.5. Right, quantification of the proportion of Olig3+ rhombic lip cells co-expressing Atoh1 between E10.5 and E14.5. (**E**) Left, immunofluorescence characterization of ventricular zone progenitor cells stained against Olig3 (red), Ptf1a (green), and the Purkinje cell marker Foxp2 (blue) at E11.5. The boxed area on the micrograph is illustrated to the right of the main photograph. Right, quantification of the proportion of Olig3+ ventricular zone cells co-expressing Ptf1a or Foxp2. The mean and SD are plotted in all graphs. n = 4 mice per age. Photomicrographs were acquired using the automatic tile scan modus (10% overlap between tiles) of the Zeiss LSM700 confocal microscope.

The online version of this article includes the following source data and figure supplement(s) for figure 1:

**Source data 1.** Source data for *Figure 1*.
**Figure supplement 1.** Expression pattern of selected bHLH factors during cerebellar development.
**Figure supplement 2.** Characterization of Olig3 expression during cerebellar development.
**Figure supplement 2—source data 1.** Source data for *Figure 1—figure supplement 2*.

Next, we characterized the expression of *Olig3* during cerebellar development by immunofluorescence. In the rhombic lip, Olig3+ cells were abundant from E10.5 to E12.5, but their numbers declined by E13.5 and were rare by E14.5 (*Figure 1B*; *Figure 1—figure supplement 2A,B*). Almost all (>98%) Olig3+ rhombic lip cells co-expressed the progenitor marker Sox2 between E10.5 and E12.5, but this co-localization, as well as the total number of Olig3+ cells, declined by E13.5 (*Figure 1C*). In addition, most (>95%) proliferative BrdU+ cells in the rhombic lip co-expressed Olig3 (*Figure 1—figure supplement 2C*). Lastly, about 30% of Olig3+ cells in the rhombic lip expressed Atoh1 (Olig3+/Atoh1+ cells; *Figure 1D*). Thus, in the rhombic lip, Olig3+ cells are progenitors, and a third of them co-express Atoh1.

In the ventricular zone, Olig3+ cells were first seen at E11.5. Their numbers peaked by E12.5 and became rare by E14.5 (*Figure 1B*; *Figure 1—figure supplement 2A,B*). Most Olig3+ cells (59%) in the ventricular zone co-expressed the progenitor marker Sox2 at E11.5 and E12.5, but this co-

**Table 1.** Categorization of bHLH transcription factors expressed or not expressed during cerebellar development in mice.

**bHLH factors expressed in cerebellar progenitor niches: Rhombic lip (RL), Ventricular zone (VZ) and/or external granule cell layer (EGL)**

| Gene name | Expressed in progenitors? | Developmental stage: embryonic (E) day | | | |
|---|---|---|---|---|---|
| | | E11.5 | E13.5 | E15.5 | E17.5/E18.5 |
| Ascl1 | Yes | VZ | VZ | Weak in VZ | Not expressed |
| Atoh1 | Yes | RL | RL and EGL | RL and EGL | RL and EGL |
| Hes1 | Yes | Not expressed | RL and VZ | Not expressed | Not expressed |
| Hes5 | Yes | RL and VZ | RL and VZ | RL and VZ | Postmitotic cells |
| Hes6 | Yes | RL and VZ | RL, VZ and EGL | EGL | No data |
| Hes7 | Yes | RL and VZ | Weak in RL, VZ | Weak in RL, VZ | Not expressed |
| Hey1 | Yes | Not expressed | Not expressed | EGL | EGL |
| Hif1a | Yes | RL and VZ | Weak in RL, VZ | Not expressed | Not expressed |
| Id1 | Yes | RL and VZ | In blood vessels | In blood vessels | In blood vessels |
| Id3 | Yes | RL and VZ | RL, VZ and EGL | RL, VZ and EGL | EGL |
| Max | Yes | Weak in RL and VZ | Not expressed | Not expressed | Not expressed |
| Mxd3 | Yes | RL and VZ | RL, VZ and EGL | EGL | EGL |
| Mxi1 | Yes | Weak in RL and VZ | RL, VZ and EGL | RL, VZ and EGL | Weak in EGL |
| Mycl | Yes | RL and VZ | RL, VZ and EGL | EGL | EGL |
| Mycn | Yes | Strong in RL and VZ | Not expressed | Not expressed | Not expressed |
| Neurod1 | Yes | Not expressed | Not expressed | Strong in EGL | Strong in EGL |
| Neurod6 | Yes | Not expressed | Weak in VZ | VZ | Broad expression |
| Neurog1 | Yes | Not expressed | VZ | Not expressed | Not expressed |
| Neurog2 | Yes | Not expressed | Weak in VZ | Strong in VZ | Not expressed |
| Olig2 | Yes | Weak in VZ | Strong in VZ | Postmitotic cells | Postmitotic cells |
| Olig3 | Yes | RL and weak in VZ | RL and VZ | Not expressed | Not expressed |
| Ptf1a | Yes | Strong in VZ | Strong in VZ | Weak in VZ | Not expressed |
| Srebf1 | Yes | Not expressed | RL, VZ and EGL | Weak in EGL | Not expressed |
| Srebf2 | Yes | Weak in RL and VZ | RL and VZ | Postmitotic cells | Postmitotic cells |
| Tcf12 | Yes | RL and VZ | RL, VZ and EGL | RL, VZ and EGL | RL, VZ and EGL |
| Tcf3 | Yes | RL and VZ | RL, VZ and EGL | RL, VZ and EGL | RL, VZ and EGL |
| Tcf4 | Yes | RL and VZ | RL, VZ and EGL | RL, VZ and EGL | RL, VZ and EGL |

*Table 1 continued on next page*

*Table 1 continued*

bHLH factors expressed in cerebellar progenitor niches: Rhombic lip (RL), Ventricular zone (VZ) and/or external granule cell layer (EGL)

| bHLH factors expressed in postmitotic cerebellar cells during development | | bHLH factors not expressed in the cerebellum during development | |
|---|---|---|---|
| Gene name: | Arntl, Arnt2, Bhlhe22, Clock, Epas1, Id2, id4, Mlx, Mnt, Mxd1, Mxd4, Myc, Ncoa1, Ncoa2, Neurod2, Neurog3, Nhlh1, Nhlh2, Npas3, Npas4, Olig1, Scx, Sim2, Usf1 and Usf2 | Gene name: | Ahr, Ahrr, Arnt, Arntl2, Ascl2, Ascl3, Ascl4, Ascl5, Atoh7, Atoh8, Bhlha15, Bhlha9, Bhlhb9, Bhlhe23, Bhlhe40, Bhlhe41, Ferd3l, Figla, Hand1, Hand2, Helt, Hes2, Hes3, Hes4, Hey2, Heyl, Hif3a, Lyl1, Mesp1, Mesp2, Mitf, Mlxip, Mlxipl, Msc, Myf5, Myf6, Myod1, Myog, Ncoa3, Neurod4, Npas1, Npas2, Sim1, Sohlh1, Sohlh2, Tal1, Tal2, Tcf15, Tcf21, Tcf23, Tcf24, Tcfl5, Tfap4, Tfe3, Tfeb, Tfec, Twist1 and Twist2 |

localization, as well as the total number of Olig3+ cells, declined by E13.5 (*Figure 1C*). Furthermore, about one-third of the BrdU+ cells in the ventricular zone co-expressed Olig3 (*Figure 1—figure supplement 2D*). Lastly, 52% of the Olig3+ ventricular zone cells co-expressed Ptf1a, while 41% co-expressed the postmitotic Purkinje cell marker Foxp2 (*Figure 1E*). This indicates that whereas most Olig3+ cells in the ventricular zone are progenitors (Olig3+/Ptf1a+/Foxp2-), Olig3 is transiently retained in early-born Purkinje cells (Olig3+/Ptf1a-/Foxp2+). We conclude that Olig3 is expressed in rhombic lip and ventricular zone progenitor cells during the generation of their earliest neuronal derivatives.

## Early derivatives from the rhombic lip and ventricular zone arise from Olig3+ progenitor cells

To obtain a complete picture of the distinct cerebellar neuron types arising from Olig3+ progenitor cells, we first carried out a long-term lineage tracing experiment. This experiment used a tamoxifen-inducible cre recombinase driven by *Olig3* (*Olig3^{creERT2/+}*) and the fluorescent reporter *Rosa26^{lsl-tdT}* that expresses a cytoplasmic Tomato fluorescent protein upon cre-mediated recombination (see the genetic strategy in *Figure 2—figure supplement 1A*). Specifically, we induced tamoxifen recombination in *Olig3^{creERT2/+}; Rosa26^{lsl-tdT/+}* mice at E10.5 and analyzed their brains by lightsheet microscopy at E19 (see *Video 1*). Three-dimensional reconstructions of recombined brains showed that Tomato+ cells were broadly distributed across the entire cerebellum of *Olig3^{creERT2/+}; Rosa26^{lsl-tdT/+}* mice (*Figure 2A–B*; *Figure 2—figure supplement 1A–A''*). In particular, we observed Tomato+ cells in the EGL, Purkinje cell layer, and dense groups of Tomato+ cells encompassing the three nuclei formed by DCN neurons: the nucleus dentatus, interpositus, and fastigii (*Figure 2A–C*). Closer inspection revealed that Tomato+ cells co-expressed Tbr1 (a marker of Fastigii DCN cells) and Brn2 (a marker of Interpositus and Dentatus DCN cells; *Figure 2D*). Thus, the distribution of cerebellar neurons with a history of *Olig3* expression suggests that Olig3+ progenitor cells generate the earliest set of ventricular zone (Purkinje cells) and rhombic lip (DCN neurons) cerebellar derivatives, and also to the later arising EGL cells from the latter progenitor domain.

We performed a second long-term lineage-tracing experiment to better define the temporal contribution of Olig3+ progenitor cells to specific cerebellar neuron types. In particular, we used *Olig3^{creERT2}* and the reporter *Mapt^{nLacZ}*, which selectively expresses a nuclear β-galactosidase (βgal) protein upon cre-mediated recombination in postmitotic (Mapt+) neurons. We

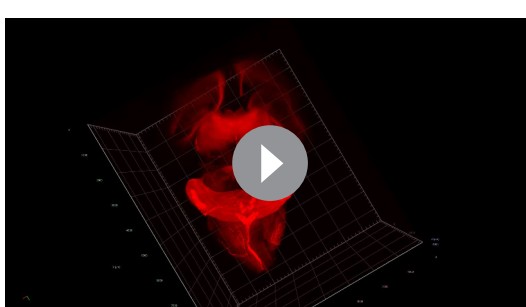

**Video 1.** Three-dimensional reconstruction of an E19 *Olig3^{creERT2/+}; Rosa26^{lsl-tdT/+}* mouse brain that was recombined at E10.5. Red fluorescence represents the somas and axons of all cells with a history of Olig3 expression. See also *Figure 2—figure supplement 1A*.
https://elifesciences.org/articles/64684#video1

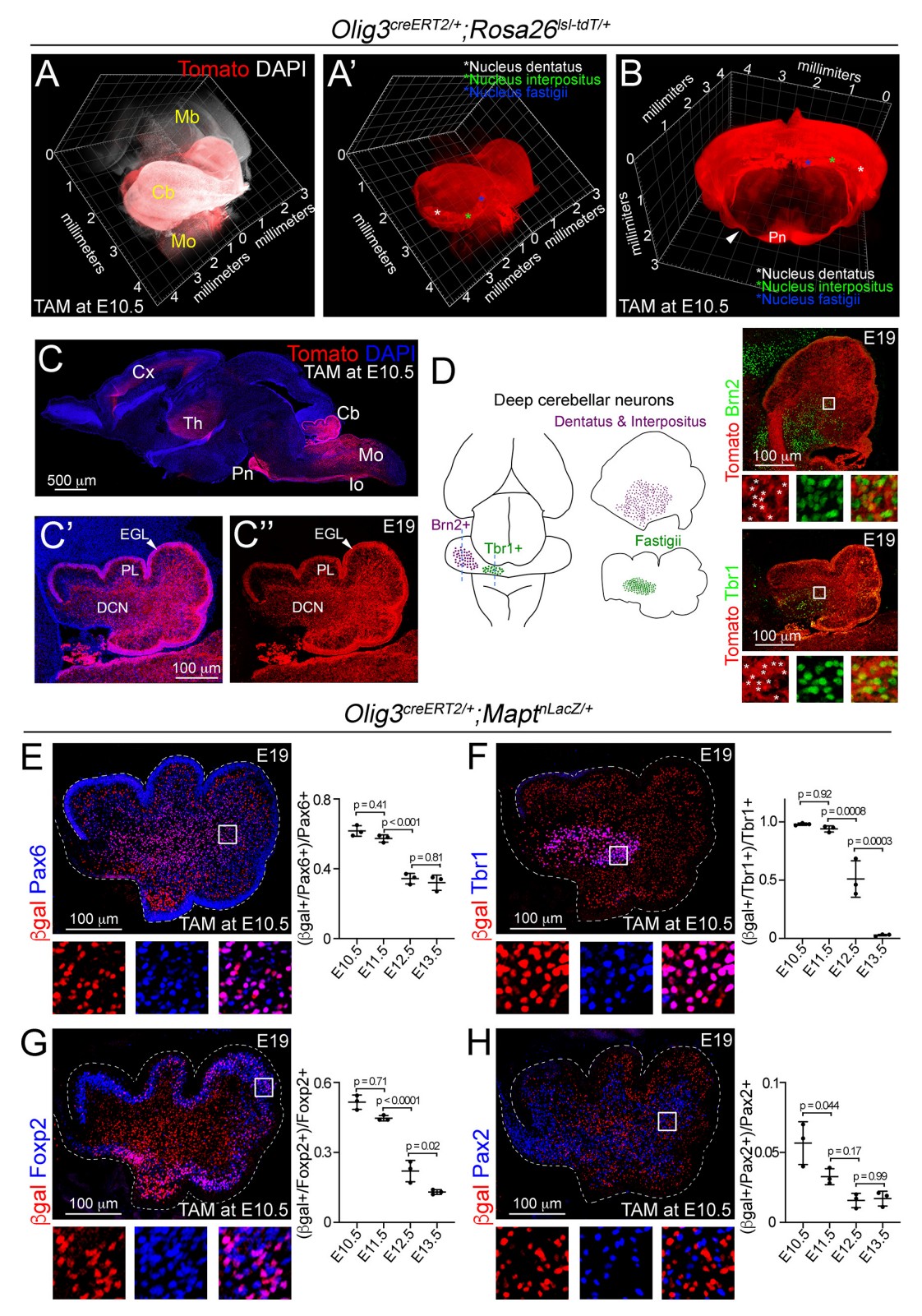

**Figure 2.** Lineage-tracing of cerebellar neurons arising from Olig3+ progenitor cells. (A–D) Analysis of Tomato+ (red) cells in *Olig3^{creERT2/+};Rosa26^{lsl-tdT/+}* mice that were recombined with tamoxifen (TAM) at E10.5 and imaged at E19. See *Figure 2—figure supplement 1A* and *Video 1* for a description of the genetic strategy and a complete reconstruction of a recombined *Olig3^{creERT2/+};Rosa26^{lsl-tdT/+}* brain. (A, A') A sagittal three-dimensional reconstruction of the cerebellum. Tomato+ cells were broadly distributed across the cerebellum and densely packed in the DCN nuclei
*Figure 2 continued on next page*

*Figure 2 continued*

(asterisks in A'). (**B**) A coronal three-dimensional reconstruction of the cerebellum. DCN nuclei are marked with asterisks. The pontine nuclei (Pn) and their axons (arrowhead), which develop from Olig3+ progenitor cells in the medulla oblongata, are labeled with Tomato. (**C**) A sagittal section stained against Tomato and DAPI (blue). Other known Olig3 derivatives, such as the thalamus (Th) including its projections to the cortex (Cx), pontine nuclei (Pn), inferior olive (Io) and many neurons in the medulla oblongata (Mo) are marked with Tomato. A magnification of the cerebellum is displayed with (**C'**) or without (**C''**) DAPI. The external granule cell layer (EGL), Purkinje cell layer (PL), and DCN neurons are labeled with Tomato. (**D**) Left, schematic display of DCN nuclei positive for Brn2 (dentatus and interpositus) and Tbr1 (fastigii). Right, sagittal cerebellar sections stained against Tomato and Brn2 or Tbr1 (green). The boxed areas are illustrated to the bottom of the main photographs displaying individual and merged fluorescent signals. Asterisks mark double positive cells. (**E–H**) Analysis of cerebellar neurons with a history of Olig3 (βgal+) expression. *Olig3*$^{creERT2/+}$;*Mapt*$^{nLacZ/+}$ mice were recombined with tamoxifen (TAM) at different embryonic stages and analyzed at E19. See *Figure 2—figure supplement 1B* for a description of the experiment. Sagittal cerebellar sections from these mice were stained against βgal (red) and markers for the EGL and granule cells (Pax6, blue in E), DCN neurons (Tbr1, blue in F), Purkinje cells (Foxp2, blue in G), and inhibitory interneurons (Pax2, blue in H). Double-positive (βgal+/marker+) cells were quantified at E19. The boxed areas on the micrographs are illustrated to the bottom of the main photographs displaying individual and merged fluorescent signals. The mean and SD are plotted in all graphs, and the dots represent the mean of individual animals. n = 3 mice per age. Significance was obtained using one-way ANOVA followed by *post hoc* Tukey's test, see *Table 2* for statistical details. Photomicrographs were acquired using the automatic tile scan modus (10% overlap between tiles) of the Zeiss LSM700 confocal microscope. The main microphotograph displayed in G was mounted on a black frame to maintain figure panel proportions.

The online version of this article includes the following source data and figure supplement(s) for figure 2:

**Source data 1.** Source data for *Figure 2*.

**Figure supplement 1.** Lineage-tracing of Olig3-derived cells.

**Figure supplement 1—source data 1.** Source data for *Figure 2—figure supplement 1*.

---

induced tamoxifen recombination in *Olig3*$^{creERT2/+}$;*Mapt*$^{nLacZ/+}$ mice at distinct embryonic stages from E10.5 to E13.5 and analyzed the cerebella of recombined mice at E19 (*Figure 2—figure supplement 1B–C*). As expected, we did not observe any βgal+ cells in the EGL, as this layer contains granule cell progenitors that do not express Mapt (arrowheads in *Figure 2—figure supplement 1C*). Both EGL and postmitotic granule cells express the transcription factor Pax6 (*Fink, 2006*; *Yeung et al., 2016*). We observed that the majority (62%) of Pax6+ postmitotic cells, outside the EGL, co-expressed βgal when recombination was induced at E10.5 or E11.5, but the proportion of Pax6+/βgal+ cells dropped when recombination was induced at later stages (*Figure 2E*). Furthermore, we found that most Tbr1+ (>99%) DCN neurons, Foxp2+ (51%) Purkinje cells, and Brn2+ (43%) DCN neurons co-expressed βgal when recombination was induced at E10.5 or E11.5, but the proportion of double-positive cells declined when recombination was induced at later stages (*Figure 2F,G*; *Figure 2—figure supplement 1D*). In contrast, few Pax2+ (6%) cells co-expressed βgal when recombination was induced at E10.5, and the number of double-positive cells was minimal when recombination was induced at later stages (*Figure 2H*). Unipolar brush cells (Tbr2+) were also observed to co-express βgal in recombined animals (*Figure 2—figure supplement 1E*). One should note that these cells derive from late rhombic lip progenitor cells at a time point (E15.5-birth) when *Olig3* is no longer expressed, which indicates that these progenitor cells, like those in the EGL, had a history of *Olig3* expression. We conclude that Olig3+ progenitor cells substantially contribute to the generation of the earliest derivatives of the ventricular zone (Purkinje cells) and rhombic lip (DCN neurons), and partially contribute to the later arising EGL cell population.

## Cerebellar hypoplasia and loss of early-born cerebellar neurons in *Olig3* mutant mice

We next analyzed the consequences of *Olig3* ablation on cerebellar development. At birth (P0), the cerebella of *Olig3* null (*Olig3*$^{-/-}$) mutant mice were drastically reduced in volume when compared to control (*Olig3*$^{+/-}$) littermates (*Figure 3A,B*). The strongest reduction in volume was observed in the medial portion of the cerebella of *Olig3* mutant mice (*Figure 3—figure supplement 1A*). Closer inspection of *Olig3* mutant cerebella revealed that they had less folia than control littermates (*Figure 3B*). Thus, ablation of *Olig3* results in severe cerebellar hypoplasia. Furthermore, the number of Tbr1+, Brn2+, Pax6+, and Foxp2+ neurons was greatly reduced in *Olig3* mutant mice (*Figure 3C–E*; *Figure 3—figure supplement 1B*). In contrast, the number of Pax2+ inhibitory interneurons increased in mutant mice (*Figure 3F*). Late born derivatives from the rhombic lip, such as Tbr2+ unipolar brush cells, were correctly specified in *Olig3* mutant mice (*Figure 3—figure*

*supplement 1C*). Together these data show that Olig3 is critically involved in cerebellar development and the generation of DCN neurons, EGL cells (including their granule cell derivatives) and Purkinje cells.

To define the developmental onset of the cerebellar deficiencies seen in *Olig3* mutant mice, we carried out a short-term lineage-tracing experiment using a knock-in mouse strain that expresses GFP from the *Olig3* locus (*Olig3$^{GFP}$*) (*Müller et al., 2005*). We compared control (*Olig3$^{GFP/+}$*) and *Olig3* mutant (*Olig3$^{GFP/GFP}$*) mice at E13.5. In control mice at this stage, DCN neurons have already migrated away from the rhombic lip and accumulated in the nuclear transitory zone, EGL cells have formed their characteristic subpial layer, and Purkinje cells have completed their specification (*Figure 3—figure supplement 2*). Compared to control littermates, E13.5 *Olig3* mutant mice showed a severe reduction in the number of DCN and EGL cells but no significant reduction in the number of Purkinje cells (*Figure 3—figure supplement 2*). Therefore, while the deficits observed in DCN neurons and EGL cells arise early during cerebellar development in *Olig3* mutant mice, the reduction in Purkinje cell numbers occurs at a later developmental stage than E13.5 (see below).

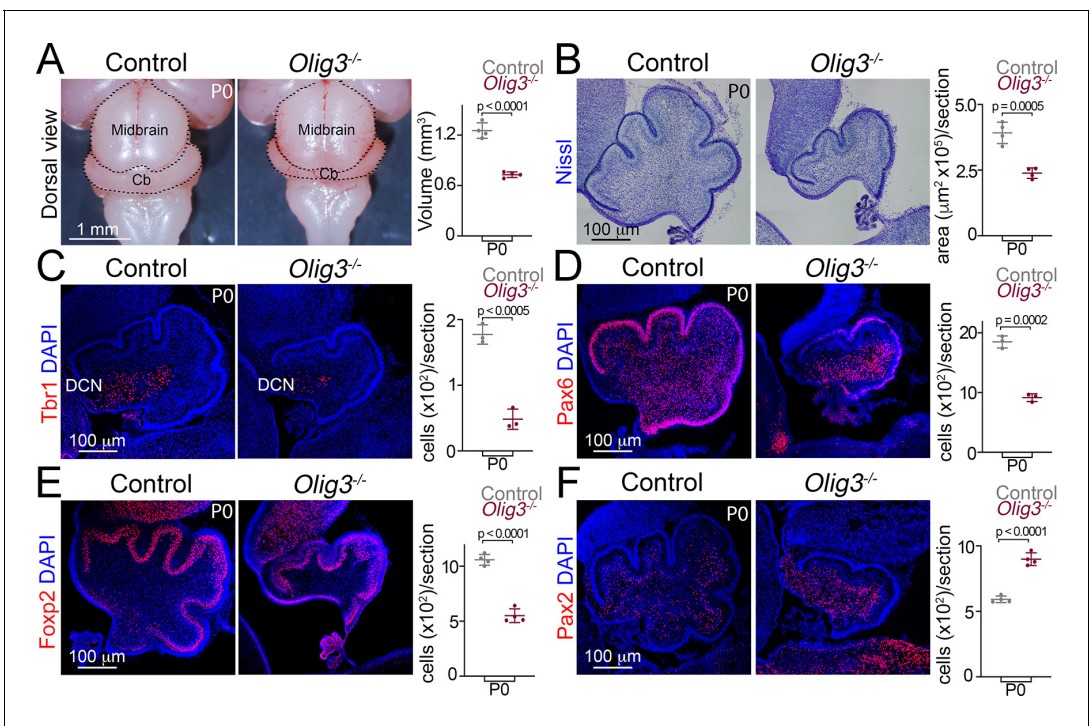

**Figure 3.** Severe cerebellar hypoplasia and neuronal loss in *Olig3* mutant mice. (**A**) Left, dorsal views of control (*Olig3$^{+/-}$*) and *Olig3* mutant (*Olig3$^{-/-}$*) cerebella at birth (**P0**). Right, quantification of cerebellar volume in newborn control and *Olig3$^{-/-}$* mice. (**B**) Left, sagittal sections of newborn control and *Olig3$^{-/-}$* cerebella stained with Nissl. Right, quantification of cerebellar area in newborn control and *Olig3$^{-/-}$* mice. (**C–F**) Immunofluorescence characterization and quantification of Tbr1+ DCN neurons (C, in red), Pax6+ EGL and granule cells (D, in red), Foxp2+ Purkinje cells (E, in red) and Pax2 + inhibitory interneurons (F, in red) in newborn control and *Olig3$^{-/-}$* mice. All cerebellar sagittal sections were counterstained with DAPI (blue). The mean and SD are plotted in all graphs, and the dots represent the mean of individual animals. n = 4 mice per genotype in A, B, E, and F; n = 3 mice per genotype in C and D. Two-tailed t-tests were performed to determine statistical significance. See *Table 2* for statistical details. Photographs in A and B were acquired with a conventional bright-field microscope and photomicrographs in C-F were acquired using the automatic tile scan modus (10% overlap between tiles) of the Zeiss LSM700 confocal microscope.

The online version of this article includes the following source data and figure supplement(s) for figure 3:

**Source data 1.** Source data for *Figure 3*.
**Figure supplement 1.** Hypoplasia and loss of defined neurons in the cerebellum of *Olig3* mutant mice.
**Figure supplement 1—source data 1.** Source data for *Figure 3—figure supplement 1*.
**Figure supplement 2.** Cerebellar neuron loss in *Olig3* mutant embryos.
**Figure supplement 2—source data 1.** Source data for *Figure 3—figure supplement 2*.
**Figure supplement 3.** Deficits of rhombic progenitor cells in *Olig3* mutant embryos.
**Figure supplement 3—source data 1.** Source data for *Figure 3—figure supplement 3*.

We next analyzed whether Atoh1 and/or Ptf1a progenitor cell numbers changed in *Olig3* mutant mice at early embryonic stages. Analysis of *Olig3* mutant mice at E11.5 and E12.5 revealed reduced numbers of Atoh1+ cells in the rhombic lip, but no change in the number of Ptf1a+ cells in the ventricular zone (*Figure 3—figure supplement 3A,B*). We then asked whether the ablation of *Olig3* might affect the proliferation of rhombic lip and ventricular zone progenitor cells and/or their viability. In the rhombic lip, the number of proliferative (BrdU+) cells was reduced in *Olig3* mutant animals (*Figure 3—figure supplement 3C*). However, no change in the number of Tunel+ apoptotic bodies (puncta) were seen at any of the analyzed embryonic stages (*Figure 3—figure supplement 3E*). In the ventricular zone of *Olig3* mutant animals neither the number of BrdU+ or Tunel+ apoptotic bodies changed (*Figure 3—figure supplement 3D,E*). Thus, the mutation of *Olig3* impairs progenitor proliferation in the rhombic lip but not in the ventricular zone, illustrating that Olig3's function in the ventricular zone differs from that in the rhombic lip.

## Ablation of *Olig3* misspecifies Purkinje cells that transform into inhibitory interneurons

We next compared the development of Foxp2+ Purkinje cells and Pax2+ inhibitory interneurons in *Olig3* mutant mice. In wildtype and heterozygous $Olig3^{GFP/+}$ mice, Pax2+ cells first appeared at E13.5 in a rostral domain of the ventricular zone that lacked expression of Olig3, and by E14.5 occupied most of the ventricular zone (*Figure 4A*). The spread of Pax2+ cells from rostral to caudal coincided with the receding of Olig3+ cells (schematically displayed in *Figure 4B*). In sharp contrast to wildtype and heterozygous $Olig3^{GFP/+}$ mice, we found supernumerary Pax2+ cells in $Olig3^{GFP/GFP}$ mutant mice from E13.5 to P0 (*Figure 4C*; *Figure 4—figure supplements 1* and *2*; quantified in *Figure 4D*). Many of the supernumerary Pax2+ cells co-expressed GFP at E13.5 and E14.5 (see magnifications in *Figure 4C* and *Figure 4—figure supplement 2A*). Thus, the ablation of *Olig3* derepresses Pax2 in the early developing cerebellum.

Surprisingly at E13.5, most Foxp2+ cells (52%) co-expressed Pax2 in *Olig3* mutant mice (*Figure 4C*, quantified in *Figure 4E*), vice versa, roughly 90% of the Pax2+ cells co-expressed Foxp2 (*Figure 4—figure supplement 2B*). Interestingly, the proportion of misspecified Foxp2+/Pax2+ (or Pax2+/Foxp2+) cells declined by E14.5 and became rare by P0 (quantified in *Figure 4E* and *Figure 4—figure supplement 2B*). The decrease of misspecified (Foxp2+/Pax2+) cells coincided with the increase of inhibitory (Foxp2-/Pax2+) interneurons seen in *Olig3* mutant animals (compare *Figure 4D* and *Figure 4E*). We thus hypothesized that misspecified (Foxp2+/Pax2+) cells in *Olig3* mutant animals might undergo a fate shift and adopt an inhibitory (Foxp2-/Pax2+) interneuron identity. To assess this hypothesis, we carried out a long-term lineage-tracing experiment using $Olig3^{creERT2}$ and the $Mapt^{nLacZ}$ reporter in an *Olig3* mutant background ($Olig3^{creERT2/GFP};Mapt^{nLacZ/+}$ mice) and analyzed βgal expression in Pax2+ inhibitory interneurons. Tamoxifen recombination in $Olig3^{creERT2/GFP};Mapt^{nLacZ/+}$ mice was induced at E10.5. We found an increase in the proportion of Pax2+/βgal+ cells in $Olig3^{creERT2/GFP};Mapt^{nLacZ/+}$ mice when compared to $Olig3^{creERT2/+};Mapt^{nLacZ/+}$ control littermates (*Figure 4F*). Notably, ectopic Pax2+/βgal+ cells in $Olig3^{creERT2/GFP};Mapt^{nLacZ/+}$ mice not only adopt an inhibitory interneuron identity (Pax2 expression), but also intermingle with Pax2+/βgal- cells underneath the Purkinje cell layer at E19 (compare inserts 1 and 2 in *Figure 4F*). Taken together, we conclude that ablation of *Olig3* in the ventricular zone results in the misspecification of Purkinje cells. These cells later change their fate and adopt an inhibitory interneuron identity (schematically displayed in *Figure 4G*).

## Olig3 cell-autonomously curtails Pax2 expression to secure Purkinje cell differentiation

To experimentally assess whether Pax2 cell-autonomously suppresses *Foxp2* expression to induce an inhibitory interneuron differentiation program, we electroporated *in utero* a Pax2-IRES-GFP-expressing vector in the ventricular zone of wildtype animals at E12.5. This is a timepoint during which Foxp2+ cells are abundant and Pax2+ cells are absent (see *Figure 5A* for a schematic display of the experimental conditions). Electroporated pCAG-Pax2-IRES-GFP (*Pax2* overexpressing) and pCAG-GFP + Empty-IRES-GFP (control) embryos were analyzed at E14.5. In pCAG-Pax2-ires-GFP electroporated mice, no GFP+/Pax2+ cell co-expressed Foxp2, whereas in pCAG-GFP + Empty-IRES-GFP electroporated mice about 72% of the GFP+ cells were also Foxp2+ (*Figure 5B*). Calcium-binding

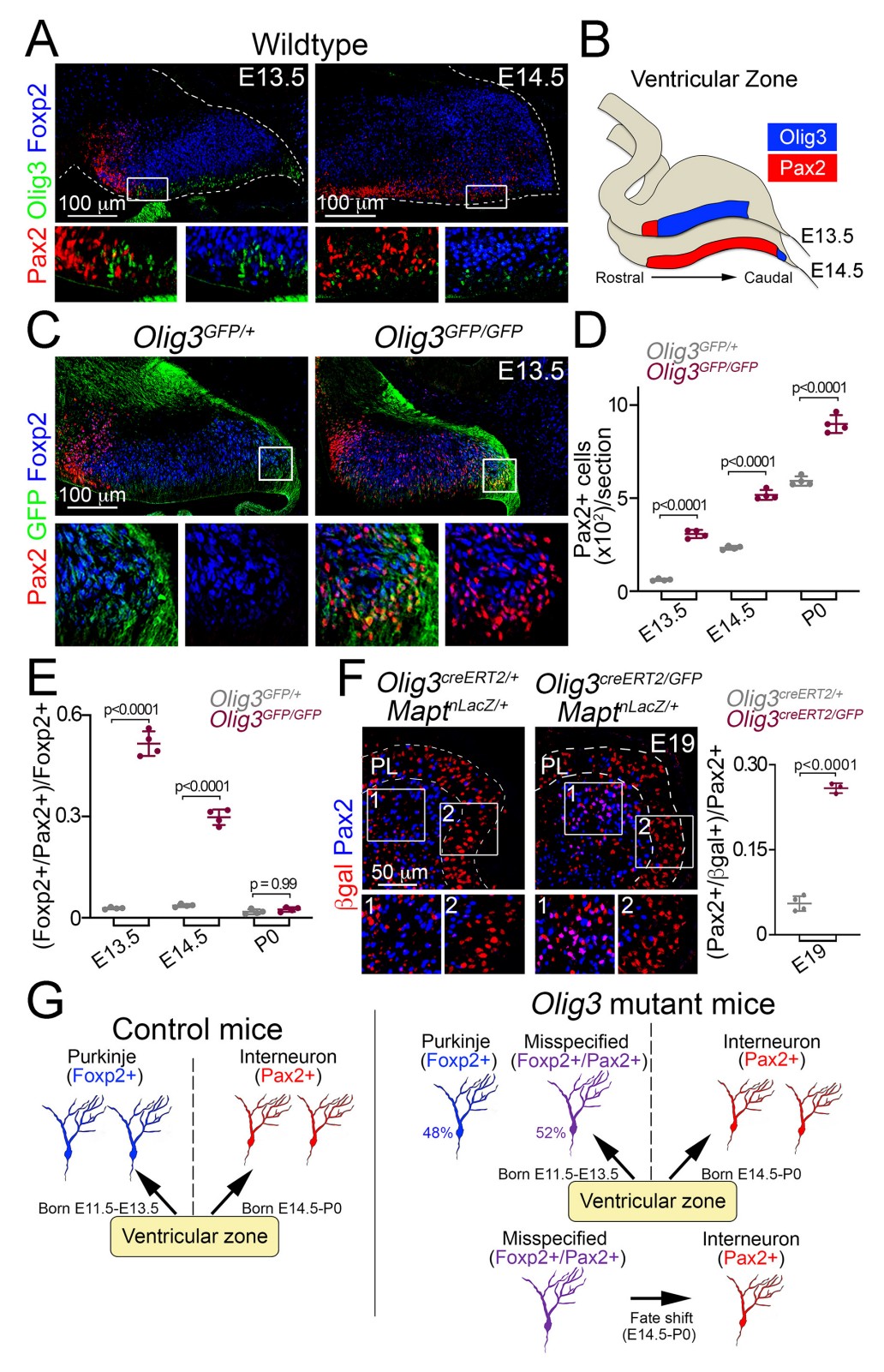

**Figure 4.** Ablation of *Olig3* misspecifies Purkinje cells which become inhibitory interneurons. (**A**) Immunofluorescence characterization of Foxp2+ (blue) Purkinje cells, Pax2+ (red) inhibitory interneurons and Olig3+ (green) progenitor cells in wildtype mice at the indicated stages. Boxed areas are magnified underneath the main photographs. (**B**) Schema illustrating the development of Pax2+ inhibitory interneurons. At E13.5, inhibitory interneurons develop in a rostral domain of the ventricular zone that lacks Olig3 expression. At E14.5, Pax2+ cells span most of the ventricular zone as
*Figure 4 continued on next page*

*Figure 4 continued*

Olig3 expression becomes extinguished. (**C**) Immunofluorescence characterization of Foxp2+ (blue) Purkinje cells and Pax2+ (red) inhibitory interneurons in E13.5 control (*Olig3^GFP/+*) and *Olig3* mutant (*Olig3^GFP/GFP*) mice. All cerebellar sagittal sections were stained against GFP (green). Boxed areas are magnified underneath the main photographs. Note that GFP+ and Foxp2+ cells ectopically express Pax2 in *Olig3^GFP/GFP* mice. See **Figure 4—figure supplement 1** for additional examples and magnifications illustrating the co-expression of Pax2 and Foxp2 in *Olig3^GFP/GFP* mice. (**D**) Quantification of Pax2+ cells in *Olig3^GFP/+* and *Olig3^GFP/GFP* mice at the indicated stages. (**E**) Quantification of the proportion of Foxp2+ Purkinje cells co-expressing Pax2+ in *Olig3^GFP/+* and *Olig3^GFP/GFP* mice at the indicated stages. (**F**) Immunofluorescence characterization and quantification of the proportion of Pax2+ (blue) inhibitory interneurons co-expressing βgal (red) in E19 control (*Olig3^creERT2/+;Mapt^nLacZ/+*) and *Olig3* mutant (*Olig3^creERT2/GFP; Mapt^nLacZ/+*) mice that were recombined at E10.5. (**G**) Schema illustrating the above findings. In control mice, the ventricular zone generates two sets of GABAergic neurons: Foxp2+ Purkinje cells (E11.5-E13.5) and Pax2+ inhibitory interneurons (E14.5- P0). In *Olig3* mutant mice, about half of the Foxp2+ cells are misspecified and co-expressed Pax2. These cells subsequently undergo a fate shift and transform into inhibitory interneurons. The mean and SD are plotted in all graphs, and the dots represent the mean of individual animals. n = 4 mice per genotype in D and E; n = 4 control mice and n = 3 *Olig3* mutant mice in F. Significance was determined using a one-way ANOVA followed by post hoc Tukey's (in D and E) or two-tailed t-test (in F) analyses, see **Table 2** for statistical details. Photomicrographs were acquired using the automatic tile scan modus (10% overlap between tiles) of the Zeiss spinning disk confocal microscope (in A and C) and the Zeiss LSM700 confocal microscope (in F).

The online version of this article includes the following source data and figure supplement(s) for figure 4:

**Source data 1.** Source data for *Figure 4*.
**Figure supplement 1.** Misspecification of Foxp2+ Purkinje cells in *Olig3* mutant mice.
**Figure supplement 2.** Misspecified Foxp2+/Pax2+ cell numbers decline over the time in *Olig3* mutant mice.
**Figure supplement 2—source data 1.** Source data for *Figure 4—figure supplement 2*.

proteins are characteristic of inhibitory interneurons in the central nervous system, among which we found Parvalbumin expression to coincide with the onset of cerebellar interneuron specification and to be virtually absent in Foxp2+ cells at E14.5 (*Figure 5—figure supplement 1A*). Next, we evaluated whether Pax2 electroporated cells acquire an inhibitory interneuron identity using the same conditions described above. In pCAG-Pax2-ires-GFP electroporated mice, a third of the Parvalbumin + cells co-expressed GFP, whereas in pCAG-GFP + Empty-IRES-GFP electroporated mice about 4% of the GFP+ cells were also Parvalbumin+ (*Figure 5—figure supplement 1B*). We conclude that Pax2 is an efficient suppressor of Foxp2 expression and that its expression seems to induce a differentiation program characteristic of inhibitory interneurons.

To assess whether Olig3 cell-autonomously suppresses *Pax2* expression, we forced the ectopic expression of *Olig3* in the ventricular zone of wildtype mice at E14, a timepoint when *Olig3* expression is almost absent and Pax2+ cells initiate their specification. Electroporated Olig3-IRES-GFP (*Olig3* overexpressing) and pCAG-GFP + Empty-IRES-GFP (control) embryos were analyzed at E15.5. The proportion of GFP+ cells that co-expressed Pax2 was greatly reduced in the ventricular zone of *Olig3*-overexpressing embryos when compared to control electroporated mice (*Figure 5C*, *Figure 5—figure supplement 2*). Thus, expression of *Olig3* is sufficient to cell-autonomously suppress Pax2 expression. We conclude that during early development, Olig3 in the ventricular zone suppresses *Pax2* in newborn Purkinje cells to prevent their misspecification and secure their identity.

## Olig3 and Olig2 specify complementary Purkinje cell populations

Our analysis of bHLH factors expressed throughout cerebellar development showed that in addition to *Olig3* and *Ptf1a*, *Ascl1* and *Olig2* are also expressed in the ventricular zone during Purkinje cell generation (*Table 1*). While ablation of *Ascl1* does not interfere with Purkinje cell development (*Grimaldi et al., 2009*; *Sudarov et al., 2011*), the exact role of Olig2 in the generation of GABAergic derivatives is unclear (*Ju et al., 2016*; *Seto et al., 2014*). In order to clarify the function of Olig2 we analyzed *Olig2* null mutant mice and found a reduction in Purkinje cell numbers and a modest increase in inhibitory interneurons (*Figure 6—figure supplement 1A,B*). We next carried out a long-term lineage-tracing experiment using *Olig2^cre* and *Mapt^nLacZ* alleles (*Olig2^cre/+;Mapt^nLacZ/+* mice) to determine the contribution of Olig2 to cerebellar GABAergic neurons. This showed that while roughly half of the Foxp2+ Purkinje cell population had a history of *Olig2* expression, few inhibitory interneurons were generated from Olig2+ progenitors (*Figure 6—figure supplement 1C*). We thus conclude that the phenotypes of *Olig3* and *Olig2* mutant mice partially overlap (summarized in *Figure 6A*; *Figure 6—figure supplement 1D*).

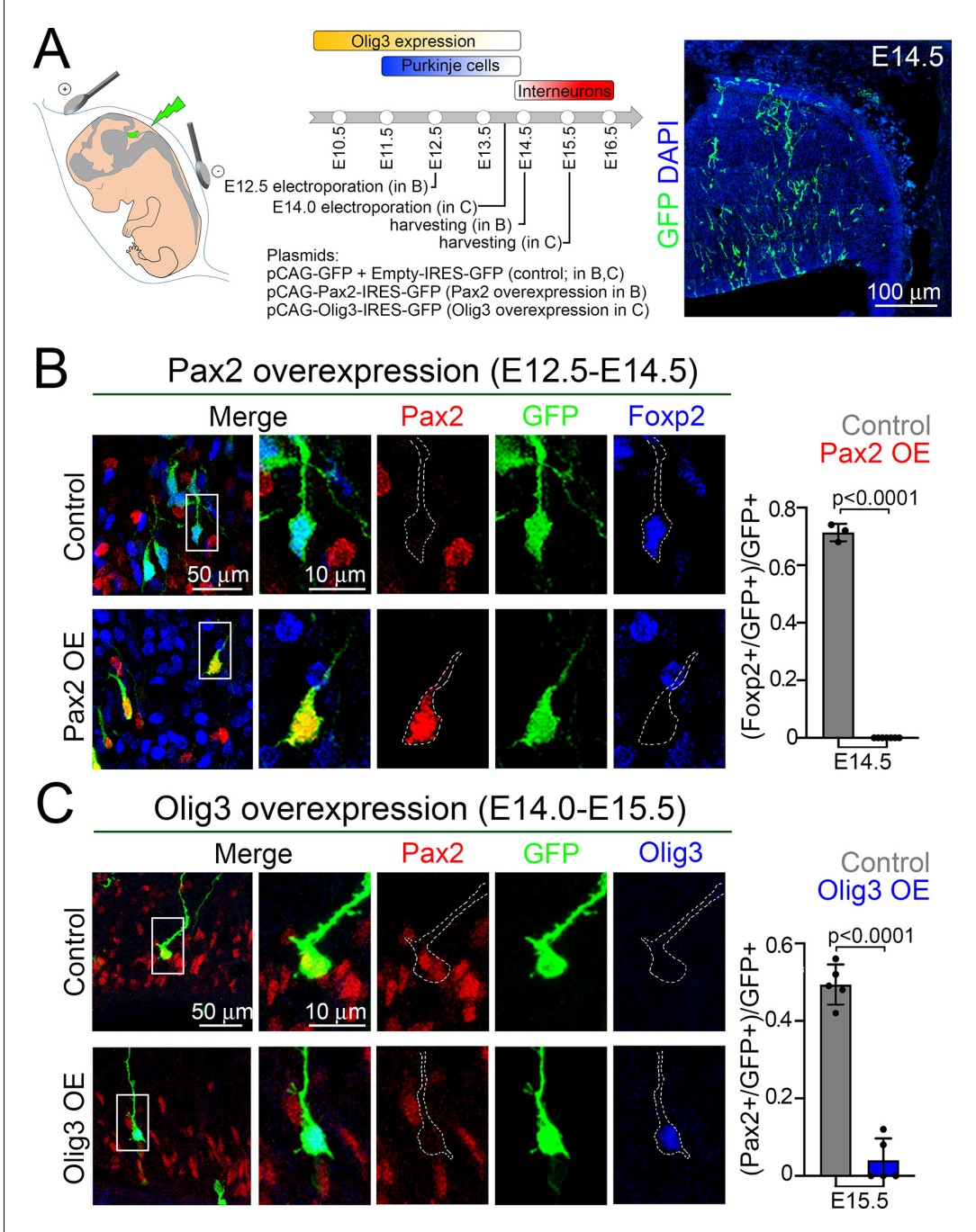

**Figure 5.** Olig3 cell-autonomously curtails *Pax2* expression to prevent the suppression of Foxp2 in newborn Purkinje cells. (**A**) Strategy to force *Olig3* and *Pax2* expression in the ventricular zone of wildtype mouse embryos. Left, illustration of the electrode position required to target the ventricular zone. Middle top, schema illustrating the temporal expression of Olig3, and the generation of Purkinje cells and inhibitory interneurons. Middle bottom, *Pax2* and *Olig3* expressing vectors were electroporated at E12.5 and E14.0, respectively. Electroporated embryos were harvested at the indicated stages. Electroporated plasmids are shown. Right, a representative cerebellar section stained with GFP (green) and DAPI (blue) of an E14.5 mouse that was electroporated with control plasmids at E12.5. (**B**) Analysis of E14.5 wildtype mice that were electroporated at E12.5 with control (pCAG-GFP + Empty-IRES-GFP) or Pax2-overexpression (pCAG-Pax2-IRES-GFP) plasmids. Left, representative analyzed cells in the cerebellum of electroporated embryos that were stained against Pax2 (red), GFP (green), and Foxp2 (blue). Right, quantification of the proportion of GFP+ cells co-expressing Foxp2 in electroporated control (Pax2-; n = 3) and Pax2-overexpressing (Pax2+; n = 7) mice. (**C**) Analysis of E15.5 wildtype mice that were electroporated at E14.0 with control (pCAG-GFP + Empty-IRES-GFP) or Olig3-overexpression (pCAG-Olig3-IRES-GFP) plasmids. Left, representative analyzed cells in the cerebellum of electroporated embryos that were stained against Pax2 (red), GFP (green) and Olig3 (blue). Right, quantification of the proportion of GFP+ cells co-expressing Pax2 in electroporated control (Olig3-; n = 5) and Olig3-overexpressing (Olig3+; n = 5) mice. See

*Figure 5 continued on next page*

*Figure 5 continued*

*Figure 5—figure supplement 2* for additional examples of electroporated cells. The mean and SD are plotted in all graphs, and the dots represent the mean of individual animals. Significance was determined using two-tailed t-tests, see *Table 2* for statistical details. Photomicrographs were manually acquired using a Leica SPL confocal microscope.

The online version of this article includes the following source data and figure supplement(s) for figure 5:

**Source data 1.** Source data for *Figure 5*.
**Figure supplement 1.** Some Pax2-electroporated cells become Parvalbumin+ interneurons.
**Figure supplement 1—source data 1.** Source data for *Figure 5—figure supplement 1*.
**Figure supplement 2.** Olig3 cell autonomously suppresses *Pax2*.

Olig3 and Olig2 are known to specify non-overlapping neuron populations during the development of the hindbrain and spinal cord (*Takebayashi et al., 2002*; *Takebayashi et al., 2000*). To determine whether Olig3 and Olig2 mark complementary ventricular zone progenitor cells that specify distinct Foxp2+ Purkinje cells, we stained the cerebella of E10.5-E14.5 wildtype embryos with antibodies against these two factors. We observed that roughly 58% of Olig3+ cells in the ventricular zone co-expressed Olig2 but not Foxp2, while the remaining 42% of the Olig3+ cells co-expressed Foxp2 but not Olig2 (*Figure 6B*). Notably, we observed no Olig3-/Olig2+ cells that co-expressed Foxp2, illustrating that differentiated Purkinje cells retain *Olig3* but not *Olig2* expression.

We then asked whether ablation of *Olig3* might affect the expression of *Olig2* in ventricular zone progenitors, and vice versa we assessed whether mutation of *Olig2* might compromise the expression of *Olig3*. Mutation of *Olig3* did not affect the expression of *Olig2*, and neither did mutation of *Olig2* affect the expression of *Olig3* in progenitor cells of the ventricular zone (*Figure 6C and D*; *Figure 6—figure supplement 1E,F*). These data demonstrate that the expression of *Olig3* and *Olig2* in ventricular zone progenitor cells is independent of the other factor. Next, we assessed whether mutation of *Olig2* might also de-repress Pax2 in newborn Purkinje cells in a similar manner as the ablation of *Olig3*. Indeed, we observed numerous Foxp2+/Pax2+ misspecified cells in *Olig2* mutant animals at E13.5 (*Figure 6E*, quantified in *Figure 6F*), but unlike in *Olig3* mutant embryos, these cells were only located in the rostral-most part of the ventricular zone (compare insets in *Figure 6E*). These data demonstrate that Olig2 specifically suppresses *Pax2* in rostrally generated Purkinje cells, while Olig3 has a broader function in the suppression of *Pax2* in most of the Purkinje cell population. We therefore conclude that Olig3 and Olig2 complementarily contribute to the correct specification of Purkinje cells by curtailing the expression of *Pax2* (schematically displayed in *Figure 6G*).

## Discussion

bHLH transcription factors are highly conserved in evolution and function as principal regulators of cell differentiation and neuronal specification (*Atchley and Fitch, 1997*; *Baker and Brown, 2018*; *Ben-Arie et al., 2000*; *Bertrand et al., 2002*; *Dennis et al., 2019*; *Dokucu et al., 1996*; *Jones, 2004*; *Sommer et al., 1996*). In this study, we sought to identify bHLH factors that regulate the specification of distinct cerebellar neuron types. We report here that Olig3 is a key player in cerebellar development and the generation of its earliest neuronal derivatives. Ablation of *Olig3* results in pronounced cerebellar hypoplasia at birth and the massive loss of DCN neurons, EGL cells including their granule cell derivatives, and Purkinje cells. These deficits are accompanied by an increase in the number of inhibitory interneurons. Our data illustrate that Olig3 regulates progenitor cell proliferation in the rhombic lip, whereas in the ventricular zone Olig3 cell-autonomously suppresses the development of inhibitory interneurons by curtailing the expression of *Pax2*. We demonstrate that Pax2 acts as an effective suppressor of the Purkinje cells differentiation program. In addition, we show that Olig3 and its close family member Olig2 specify complementary Purkinje cell populations.

Here, we show that Olig3 is critically involved in the generation of EGL cells as well as DCN neurons. Earlier studies revealed that these rhombic lip derivatives depend on Atoh1 for their development, as loss of Atoh1 results in the severe reduction of EGL cells and impairs the development of DCN neurons (*Ben-Arie et al., 1997*; *Gazit et al., 2004*; *Jensen, 2004*; *Machold and Fishell, 2005*; *Machold et al., 2011*; *Wang et al., 2005*; *Yamada et al., 2014*). Our long-term lineage-tracing studies demonstrated that most EGL and DCN cells have a history of *Olig3* expression, ablation of which massively reduced their cell numbers. In the early rhombic lip (E10.5-E13.5), we found that most

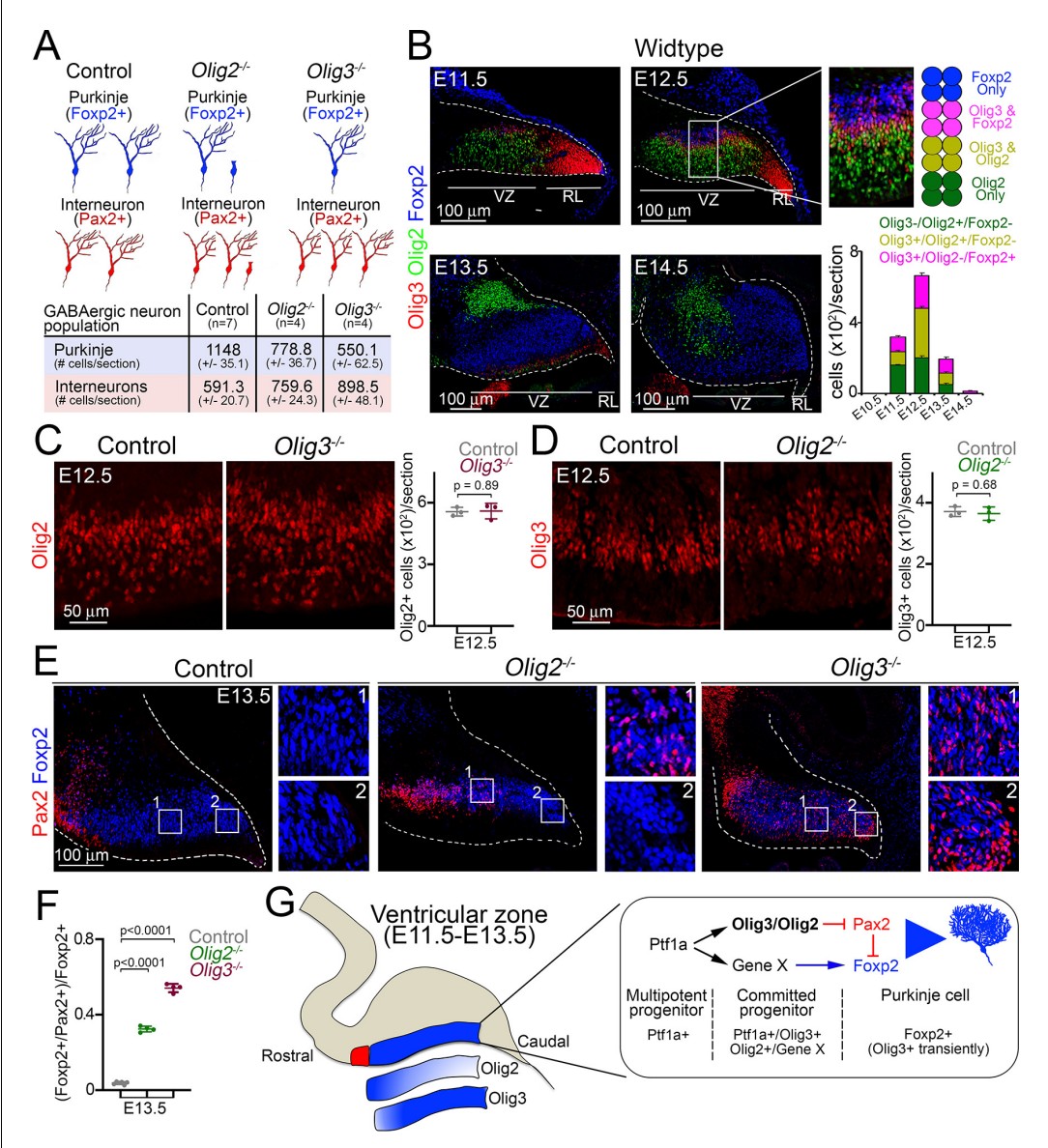

**Figure 6.** Complementary functions of Olig3 and Olig2 during Purkinje cell development. (**A**) Schema and quantification of the phenotypes observed in *Olig2* and *Olig3* mutant mice with respect to the development of GABAergic cerebellar neurons. See also *Figure 3E and F* (*Olig3* mutant analysis) and *Figure 6—figure supplement 1A and B*. (*Olig2* mutant analysis). n numbers are indicated in the brackets. (**B**) Immunofluorescence characterization and quantification of Olig3+ (red), Olig2+ (green), and Foxp2+ (blue) cells in the ventricular zone at indicated embryonic stages (n = 4 mice per age). (**C**) Immunofluorescence characterization and quantification of Olig2+ (red) cells in the ventricular zone of *Olig3* (*Olig3⁻/⁻*) mutant mice at E12.5 (n = 3 mice per genotype). See also *Figure 6—figure supplement 1E* (**D**) Immunofluorescence characterization and quantification of Olig3+ (red) cells in the ventricular zone of *Olig2* (*Olig2⁻/⁻*) mutant mice at E12.5 (n = 3 mice per genotype). See also *Figure 6—figure supplement 1F*. (**E**) Immunofluorescence comparison of Foxp2+ (blue) Purkinje cells and Pax2+ (red) inhibitory neurons in control versus *Olig2⁻/⁻* and *Olig3⁻/⁻* mutant mice at E13.5. Numbered boxed areas are displayed to the right of the main photographs. (**F**) Quantification of the proportion of Foxp2+ cells co-expressing Pax2 in control (n = 6), *Olig2⁻/⁻* (n = 4), and *Olig3⁻/⁻* (n = 4) mutant mice at E13.5. (**G**) Schematic summary explaining the function of Olig3 and Olig2 in the ventricular zone during the specification of Purkinje cells. Induction of Olig3 and Olig2, in committed Ptf1a+ progenitor cells, curtails the expression of Pax2 to allow for the correct specification of Purkinje cells. Olig2 predominantly operates in the rostral ventricular zone, whereas Olig3 has a broader function and is transiently retained in newborn Purkinje cells. The suppression of Pax2 is critical for Purkinje cell development, as it can override the Purkinje cell differentiation program. The mean and SD are plotted in all graphs, and the dots represent the mean of individual animals. Significance was determined using a one-way ANOVA followed by post hoc Tukey (in F) or two-tailed t-test (in C and D) analyses, see *Table 2* for statistical details. Photomicrographs were acquired using the automatic tile scan modus (10% overlap between tiles) of the Zeiss spinning disk confocal microscope (in C-E) and the Zeiss LSM700 confocal microscope (in B).

*Figure 6 continued on next page*

*Figure 6 continued*

The online version of this article includes the following source data and figure supplement(s) for figure 6:

**Source data 1.** Source data for *Figure 6*.
**Figure supplement 1.** Analysis of GABAergic neurons in O*lig2* mutant mice.
**Figure supplement 1—source data 1.** Source data for *Figure 6—figure supplement 1*.

proliferative progenitor cells (Sox2+/BrdU+) co-expressed Olig3 and a third of them co-expressed Atoh1 (Olig3+/Atoh1+ cells). This temporal window overlaps with the generation of DCN cells (*Fink, 2006*; *Sekerková et al., 2004*; *Wang et al., 2005*; *Yamada et al., 2014*), which are the most reduced neuron type in *Olig3* mutant mice (this study). Thus, Olig3 is essential for DCN neuron development. Ablation of *Olig3* reduced the number of BrdU+ (proliferative) progenitor cells in the rhombic lip, and consequently decreased the number of Atoh1+ cells. This impairment led to smaller numbers of EGL cells and, therefore, to fewer differentiated granule cells. The severe loss of EGL cells and their granule cell derivatives seems to largely account for the pronounced cerebellar hypoplasia observed in *Olig3* mutant mice, and has also been observed after the loss of EGL cells in other studies (*Ben-Arie et al., 1997*). One should note, however, that the reduced numbers of granule cells in *Olig3* mutant mice might be mainly independent of Olig3 and due to the reduction of instructive signals emanating from Purkinje cells, which are severely affected in these mutant mice. Indeed, available evidence shows that Purkinje cells regulate EGL proliferation and the differentiation of granule cells via sonic hedgehog signaling (*Dahmane and Ruiz i Altaba, 1999*; *Wallace, 1999*; *Wechsler-Reya and Scott, 1999*).

During the development of rhombomeres 2–7, there exists a dorsal progenitor domain (called dA1) that also co-expresses Olig3 and Atoh1 (*Hernandez-Miranda et al., 2017a*; *Liu et al., 2008*; *Storm et al., 2009*). This domain generates the mossy fiber precerebellar (pontine, lateral reticular, external cuneate) nuclei. Like in the rhombic lip, ablation of *Olig3* greatly reduces the number of Atoh1+ cells in this area and their derivatives (*Liu et al., 2008*; *Storm et al., 2009*). This phenomenon also occurs in the spinal cord when *Olig3* is ablated (*Müller et al., 2005*). Thus, Olig3 has a conserved function in the proliferation of Atoh1+ progenitor cells.

We also show that the ablation of *Olig3* results in the development of supernumerary inhibitory interneurons. Both Purkinje cells and inhibitory interneurons depend on Ptf1a for their development (*Hashimoto and Mikoshiba, 2003*; *Hoshino et al., 2005*; *Leto et al., 2006*; *Yamada et al., 2014*). We predominantly found expression of Olig3 in the ventricular zone between E11.5 and E13.5, the temporal window during which Purkinje cells are specified. In the ventricular zone, about half of the Olig3+ cells co-express Ptf1a, while the rest co-express the Purkinje cell marker Foxp2. This shows that Olig3 expression is initiated in progenitors and transiently retained in newborn Purkinje cells. Ablation of *Olig3* neither impaired the number of Ptf1a+ cells nor their proliferation. Strikingly, around half of newborn Purkinje cells erroneously co-expressed Pax2 in *Olig3* mutant mice at E13.5, illustrating that ablation of *Olig3* misspecifies newborn Purkinje cells. In *Olig3* mutant mice, the number of misspecified cells declined over time and became rare by P0. This decline correlated with a parallel increase in the number of inhibitory interneurons. Thus, the primary function of Olig3 in the ventricular zone is to secure the development of Purkinje cells by cell-autonomously suppressing an alternative program that specifies inhibitory interneurons. In this context, our functional data demonstrate that forced expression of *Olig3*, during the temporal generation of inhibitory interneurons, is sufficient to curtail *Pax2* expression. Furthermore, our functional data demonstrate that Pax2 acts as an effective suppressor of Foxp2 and the Purkinje differentiation cell program. In agreement with our findings, it was previously shown that supernumerary inhibitory neurons become specified at the expense of excitatory neurons in the hindbrain and spinal cord of *Olig3* mutant mice (*Müller et al., 2005*; *Storm et al., 2009*; *Zechner et al., 2007*). Interestingly, there is a unique progenitor domain in rhombomere 7 (called dA4) that co-expresses Olig3 and Ptf1a, which generates the pre-cerebellar climbing fiber neurons of the inferior olive (reviewed in *Hernandez-Miranda et al., 2017a*). In the absence of Olig3, inferior olive neurons and many spinal cord excitatory neurons seem to change their fate and erroneously adopt an inhibitory interneuron identity (*Liu et al., 2008*; *Müller et al., 2005*; *Storm et al., 2009*). This suggests that inhibitory interneurons are the default neuronal type generated from the brainstem, spinal cord and the ventricular zone of the cerebellum.

Based on short-term lineage-tracing experiments, *Seto et al., 2014* postulated a 'temporal identity transition' model in which Olig2+ Purkinje cell progenitors transition into inhibitory interneuron progenitors (*Seto et al., 2014*). From this model, one would expect that inhibitory interneurons would have a history of *Olig2* expression. In keeping with observations made by *Ju et al., 2016*, our long-term lineage-tracing experiments using *Olig2^cre^* and *Olig3^creERT2^* mice showed that Pax2+ inhibitory interneurons rarely have a history of *Olig2* or *Olig3* expression. This casts doubt on the 'temporal identity transition' model as both factors are abundantly expressed in Ptf1a+ progenitors during the specification of Purkinje cells (this work and *Seto et al., 2014*). Our data unambiguously show that neither Olig2 nor Olig3 control the transition of early (Purkinje) to late (inhibitory interneuron) ventricular zone progenitor cells. Rather, our work demonstrates that these factors are essential for the correct specification of Purkinje cells by curtailing an inhibitory interneuron transcriptional program.

Development of the central nervous system is characterized by molecular 'grids' of combinatorial transcription factor expression that single out distinct progenitor domains. It is from here that the enormous diversity of neuron types is generated (reviewed in *Alaynick et al., 2011*; *Hernandez-Miranda et al., 2017a*; *Hernández-Miranda et al., 2010*; *Jessell, 2000*). Here, we show that cerebellar DCN neurons and internal granule cells develop from Olig3+/Atoh1+ rhombic lip progenitor cells, whereas Purkinje cells derive from Olig3+/Ptf1a+ ventricular zone progenitors. In the mature cerebellum, DCN neurons and granule cells receive input from brainstem precerebellar mossy fiber neurons that originate from progenitor cells that co-express Olig3 and Atoh1, whilst Purkinje cells receive input from climbing fiber neurons that emerge from progenitors that co-express Olig3 and Ptf1a (*Liu et al., 2008*; *Storm et al., 2009*). The question of how these progenitor cells, located at such distant positions, acquire similar molecular signatures to specify both targets and inputs that in turn form functional cerebellar circuits remains to be elucidated.

## Materials and methods

### Key resources table

| Reagent type (species) or resource | Designation | Source or reference | Identifiers | Additional information |
|---|---|---|---|---|
| Strain, strain background (*M. musculus*) | *Olig3^CreERT2^* | *Storm et al., 2009*. | RRID:MGI:3833734 | |
| Strain, strain background (*M. musculus*) | *Olig3^GFP^* | *Müller et al., 2005*. | | |
| Strain, strain background (*M. musculus*) | *Mapt^nLacZ^* | *Hippenmeyer et al., 2005*. | The Jackson Laboratory, Stock No: 021162 | |
| Strain, strain background (*M. musculus*) | *Rosa26^lsl-tdT^ (Ai14)* | *Madisen et al., 2010*. | The Jackson Laboratory, Stock No. 007908 | |
| Strain, strain background (*M. musculus*) | *Olig2^Cre^* | *Dessaud et al., 2007*. | | |
| Antibody | Anti-β-gal (Chicken polyclonal) | Abcam | ab9361 RRID:AB_307210 | (1:1,000) |
| Antibody | Anti-GFP (Chicken polyclonal) | Abcam | ab13970 RRID:AB_300798 | (1:500) |
| Antibody | Anti-Brn2 (Goat polyclonal) | Abcam | ab101726 RRID:AB_10710183 | (1:1,000) |
| Antibody | Anti-Foxp2 (Goat polyclonal) | Abcam | ab58599 RRID:AB_941649 | (1:1,000) |
| Antibody | Anti-Olig3 (Guinea pig polyclonal) | Gift from T. Muller | Homemade | (1:5,000) |

*Continued on next page*

*Continued*

| Reagent type (species) or resource | Designation | Source or reference | Identifiers | Additional information |
|---|---|---|---|---|
| Antibody | Anti-Foxp2 (Rabbit polyclonal) | Abcam | ab16046 RRID:AB_2107107 | (1:1,000) |
| Antibody | Anti-GFP (Rabbit polyclonal) | Abcam | ab290 RRID:AB_303395 | (1:500) |
| Antibody | Anti-Pax2 (Rabbit monoclonal) | Abcam | EP3251 RRID:AB_1603338 | (1:1,000) |
| Antibody | Anti-Sox2 (Rabbit polyclonal) | Abcam | ab97959 RRID:AB_2341193 | (1:1,000) |
| Antibody | Anti-Tbr1 (Rabbit polyclonal) | Abcam | ab31940 RRID:AB_2200219 | (1:1,000) |
| Antibody | Anti-Tbr2 (Rabbit polyclonal) | Abcam | ab23345 RRID:AB_778267 | (1:1,000) |
| Antibody | Anti-Ptf1a (Rabbit polyclonal) | Gift from J. Johnson | Homemade | (1:5,000) |
| Antibody | Anti-Olig2 (Rabbit polyclonal) | Merck Millipore | AB9610 RRID:AB_570666 | (1:1,000) |
| Antibody | Anti-Pax6 (Rabbit polyclonal) | Merck Millipore | AB2237 RRID:AB_1587367 | (1:1,000) |
| Antibody | Anti-RFP (Rabbit polyclonal) | Rockland | 600-401-379 RRID:AB_2209751 | (1:500) |
| Antibody | Anti-Caspase-3 (Rabbit polyclonal) | R and D Systems | AF835 RRID:AB_2243952 | (1:1,000) |
| Antibody | Anti-Parvalbumin (Rabbit polyclonal) | Swant | PV 27 RRID:AB_2631173 | (1:3,000) |
| Antibody | Anti-Atoh1 (Rabbit polyclonal) | Gift from T. Jessell | Homemade | (1:10,000) |
| Antibody | Anti-BrdU (Rat monoclonal) | Abcam | ab6326 RRID:AB_305426 | (1:2,000) |
| Antibody | Anti-GFP (Rat monoclonal) | Nacalai Tesque | GF090R RRID:AB_10013361 | (1:2,000) |
| Antibody | Donkey anti-species Alexa Fluor 488/568/647 | Jackson ImmunoResearch | Various | (1:500) |
| Commercial Assay or kit | BrdU | Sigma-Aldrich | B5002-1G | 16 mg/ml in 0.9% saline solution |
| Recombinant DNA reagent | pCAG-GFP | Addgene | Plasmid #11150 RRID:Addgene_11150 | |
| Recombinant DNA reagent | pCAG-Empty-IRES-GFP | This paper | | |
| Recombinant DNA reagent | pCAG-Olig3-IRES-GFP | This paper | | vector: pCAG; cDNA fragment: mouse *Olig3* |
| Recombinant DNA reagent | pCAG-Pax2-IRES-GFP | This paper | | vector: pCAG; cDNA fragment: mouse *Pax2* |
| Sequence-based reagent (*M. musculus*) | Mouse *Olig3* forward primer | Olig3FW | PCR Primer | ATGAATTCTGATTCGAGC |
| Sequence-based reagent (*M. musculus*) | Mouse *Olig3* reverse primer | Olig3RV | PCR Primer | TTAAACCTTATCGTCGTC |
| Sequence-based reagent (*M. musculus*) | Mouse *Pax2* forward primer | Pax2FW | PCR Primer | ATGGATATGCACTGCAAAGCAG |
| Sequence-based reagent (*M. musculus*) | Mouse *Pax2* reverse primer | Pax2RV | PCR Primer | GTGGCGGTCATAGGCAGC |

*Continued on next page*

*Continued*

| Reagent type (species) or resource | Designation | Source or reference | Identifiers | Additional information |
|---|---|---|---|---|
| Software, algorithm | GraphPad Prism | GraphPad Software | RRID:SCR_002798 | Prism 8 |
| Software, algorithm | Adobe Photoshop | Adobe | RRID:SCR_014199 | Adobe Photoshop CS6 |
| Software, algorithm | ImageJ | NIH | RRID:SCR_002285 | |
| Software, algorithm | Arivis Vision4D | Arivis | RRID:SCR_018000 | Arivis Vision4D 3.2 |

## Animals

All animal experimental procedures were done in accordance to the guidance and policies of the Charite Universitatsmedizin, Berlin, Germany; Max-Delbrück-Center for Molecular Medicine, Berlin, Germany; and the Institute of Neuroscience, Lobachevsky University of Nizhny Novgorod, Russian Federation. Mouse strains used for this study were: *Olig3creERT2* (*Storm et al., 2009*), *Olig3GFP* (*Müller et al., 2005*), *MaptnLacZ* (*Hippenmeyer et al., 2005*), *Rosa26lsl-tdT* (*Madisen et al., 2010*), and *Olig2cre* (*Dessaud et al., 2007*). All strains were maintained in a mixed genetic background.

For tamoxifen treatment, pregnant dams were treated with tamoxifen (Sigma-Aldrich; 20 mg/ml dissolved in sunflower oil) as described previously (*Hernandez-Miranda et al., 2017b*; *Storm et al., 2009*). Tamoxifen delays labor in rodents and humans (*Lizen et al., 2015*). Therefore, offspring from tamoxifen-treated dams were delivered by caesarean section at E19.

## Histology and cell quantifications

Immunofluorescence and tissue processing were performed as previously described (*Hernández-Miranda et al., 2011*). Briefly, mouse tissue (E10.5–P0) was fixed in 4% paraformaldehyde (PFA), made in phosphate buffered saline (PBS), for 3 hr at 4°C. After fixation, brains were cryoprotected overnight in 30% sucrose in PBS, embedded and frozen in Tissue-Tek OCT (Sakura Finetek), and sectioned at 20 µm using a cryostat. Sections were washed in PBS and blocked in PBS containing 5% normal goat serum (Sigma-Aldrich) (v/v) and 0.1% Triton X-100 (v/v) (Sigma-Aldrich) at room temperature for 1 hr. They were subsequently incubated in primary antibodies at room temperature overnight. After incubation in primary antibodies, sections were washed in PBS and then incubated in secondary antibodies for 2 hr at room temperature. Primary and secondary antibodies used in this study are displayed in the Key Resources Table. For a 45 min BrdU pulse labeling, BrdU (Sigma-Aldrich) was diluted to a concentration of 16 mg/ml in saline solution and injected intraperitoneally.

Cell quantifications were performed in a non-blind manner on non-consecutive 20-µm-thick brain sections encompassing the complete lateral-medial cerebellar axis. On average six to ten sections per animal were used for quantifications. E12.5 whole-mount embryos were analysed for β-gal activity with X-gal (0.6 mg/ml; Merck Millipore, B4252) in PBS buffer containing 4 mM potassium ferricyanide, 4 mM potassium ferrocyanide, 0.02% NP-40 and 2 mM MgCl2 as previously described (*Comai and Tajbakhsh, 2014*). For the estimation of the cerebellar volume and area, consecutive 20-µm-thick sagittal sections were collected encompassing the whole cerebellum and stained with Nissl. Roughly 32–35 sections of the cerebellum were obtained per animal (four animals/genotype). The area of every section was measured using ImageJ; NIH, version 1.34 n. Estimation of the total volume of the cerebellum was obtained by application of Cavalieri's method (*West, 2012*). Fluorescence images were acquired using: (i) a Zeiss LSM 700 confocal microscope using the automatic tile scan modus (10% overlap between tiles) and assembled using ZEN2012, (ii) a Zeiss spinning disk confocal microscope using the automatic tile scan modus (10% overlap between tiles) and assembled using ZEN2012, and (iii) a Leica SPL confocal microscope. Photographs obtained with the Leica SPL confocal microscope were manually acquired and these were assembled using Image J. Unless otherwise specified all photomicrographs were acquired in a non-blind manner.

## Brain clearing, lightsheet microscopy and analysis

Brains were cleared using the CUBIC protocol (*Susaki et al., 2015*). Briefly, brains were dissected and fixed overnight at 4°C in 4% paraformaldehyde made in PBS. After washing overnight in PBS, lipids were removed using Reagent-1 (25% urea, 25% Quadrol, 15% Triton X-100, 35% dH$_2$O) at 37°C until brains were transparent (4 days). The brains were then washed overnight at 4°C in PBS to

**Table 2.** Description of the statistical analyses used in this study.

| Fig. | N | Descriptive statistics | Test used | p Value | Degrees of freedom and F/t/z/R/ETC | Pos hoc analysis | Adjusted p value |
|---|---|---|---|---|---|---|---|
| 2E | Three mice (E10.5) three mice (E11.5) three mice (E12.5) three mice (E13.5) | Mean and SD | Ordinary one-way ANOVA | <0.0001 | F: 67.20 F(DFn, DFd): 0.1441 (3, 8) | Tukey's multiple comparative test | As indicated in the figure |
| 2F | Three mice (E10.5) three mice (E11.5) three mice (E12.5) three mice (E13.5) | Mean and SD | Ordinary one-way ANOVA | <0.0001 | F: 93.57 F(DFn, DFd): 1.868 (3, 8) | Tukey's multiple comparative test | As indicated in the figure |
| 2G | Three mice (E10.5) three mice (E11.5) three mice (E12.5) three mice (E13.5) | Mean and SD | Ordinary one-way ANOVA | <0.0001 | F: 122.2 F(DFn, DFd): 1.096 (3, 8) | Tukey's multiple comparative test | As indicated in the figure |
| 2H | Three mice (E10.5) three mice (E11.5) three mice (E12.5) three mice (E13.5) | Mean and SD | Ordinary one-way ANOVA | <0.0001 | F: 13.65 F(DFn, DFd): 0.8851 (3, 8) | Tukey's multiple comparative test | As indicated in the figure |
| 3A | Four control mice four mutant mice | Mean and SD | Unpaired t-test (two-tailed) | <0.0001 | t = 15.13; df = 6 | - | - |
| 3B | Four control mice four mutant mice | Mean and SD | Unpaired t-test (two-tailed) | 0.0005 | t = 6.742; df = 6 | - | - |
| 3C | Three control mice three mutant mice | Mean and SD | Unpaired t-test (two-tailed) | 0.0005 | t = 10.47; df = 4 | - | - |
| 3D | Three control mice three mutant mice | Mean and SD | Unpaired t-test (two-tailed) | 0.0002 | t = 13.45; df = 4 | - | - |
| 3E | Four control mice four mutant mice | Mean and SD | Unpaired t-test (two-tailed) | <0.0001 | t = 16.89; df = 6 | - | - |
| 3F | Four control mice four mutant mice | Mean and SD | Unpaired t-test (two-tailed) | <0.0001 | t = 11.26; df = 6 | - | - |
| 4D | Four control (E13.5) four mutant (E13.5) four control (E14.5) four mutant (E14.5) four control (P0) four mutant (P0) | Mean and SD | Ordinary one-way ANOVA | <0.0001 | F: 498.6 F(DFn, DFd): 1.252 (5, 18) | Tukey's multiple comparative test | As indicated in the figure |
| 4E | Four control (E13.5) four mutant (E13.5) four control (E14.5) four mutant (E14.5) four control (P0) four mutant (P0) | Mean and SD | Ordinary one-way ANOVA | <0.0001 | F: 514.2 F(DFn, DFd): 7.873 (5, 18) | Tukey's multiple comparative test | As indicated in the figure |
| 4F | Four control mice three mutant mice | Mean and SD | Unpaired t-test (two-tailed) | <0.0001 | t = 22.92; df = 5 | - | - |
| 5B | Three control-OE mice 7 Pax2-OE mice | Mean and SD | Unpaired t-test (two-tailed) | <0.0001 | t = 67.67; df = 8 | - | - |
| 5C | Five control-OE mice 5 Olig3-OE mice | Mean and SD | Unpaired t-test (two-tailed) | 0.0001 | t = 13.24; df = 8 | - | - |
| 6C | Three control mice three mutant mice | Mean and SD | Unpaired t-test (two-tailed) | 0.8969 | t = 0.1380; df = 4 | - | - |
| 6D | Three control mice three mutant mice | Mean and SD | Unpaired t-test (two-tailed) | 0.6835 | t = 0.4387; df = 4 | - | - |

*Table 2 continued on next page*

*Table 2 continued*

| Fig. | N | Descriptive statistics | Test used | p Value | Degrees of freedom and F/t/z/R/ETC | Pos hoc analysis | Adjusted p value |
|------|---|------------------------|-----------|---------|-------------------------------------|------------------|------------------|
| 6F | Four control<br>4 Olig3 mutant<br>4 Olig2 mutant | Mean and SD | Ordinary one-way ANOVA | <0.0001 | F: 1416<br>F(DFn, DFd): 0.8767 (2, 11) | Tukey's multiple comparative test | As indicated in the figure |
| 2-fs 1C | Three mice (E10.5)<br>three mice (E11.5)<br>three mice (E12.5)<br>three mice (E13.5) | Mean and SD | Ordinary one-way ANOVA | <0.0001 | F: 299.8<br>F(DFn, DFd): 0.3133 (3, 8) | Tukey's multiple comparative test | As indicated in the figure |
| 2-fs 1D | Three mice (E10.5)<br>three mice (E11.5)<br>three mice (E12.5)<br>three mice (E13.5) | Mean and SD | Ordinary one-way ANOVA | <0.0001 | F: 100.9<br>F(DFn, DFd): 0.2601 (3, 8) | Tukey's multiple comparative test | As indicated in the figure |
| 2-fs 1E | Three mice (E10.5)<br>three mice (E11.5)<br>three mice (E12.5)<br>three mice (E13.5) | Mean and SD | Ordinary one-way ANOVA | <0.0001 | F: 85.64<br>F(DFn, DFd): 0.08333 (3, 8) | Tukey's multiple comparative test | As indicated in the figure |
| 3-fs 1B | Three control mice<br>three mutant mice | Mean and SD | Unpaired t-test (two-tailed) | 0.0093 | t = 4.707; df = 4 | - | - |
| 3-fs 1C | Three control mice<br>three mutant mice | Mean and SD | Unpaired t-test (two-tailed) | 0.2799 | t = 1.249; df = 4 | - | - |
| 3-fs 2A | Three control mice<br>three mutant mice | Mean and SD | Unpaired t-test (two-tailed) | <0.0001 | t = 16.84; df = 4 | - | - |
| 3-fs 2B | Three control mice<br>three mutant mice | Mean and SD | Unpaired t-test (two-tailed) | <0.0001 | t = 17.51; df = 4 | - | - |
| 3-fs 2C | Three control mice<br>three mutant mice | Mean and SD | Unpaired t-test (two-tailed) | 0.0801 | t = 2.331; df = 4 | - | - |
| 3-fs 3A | Four control (E11.5)<br>four mutant (E11.5)<br>four control (E12.5)<br>four mutant (E12.5) | Mean and SD | Ordinary one-way ANOVA | <0.0001 | F: 139.7<br>F(DFn, DFd): 2.328 (3, 12) | Tukey's multiple comparative test | As indicated in the figure |
| 3-fs 3B | Three control (E11.5)<br>four mutant (E11.5)<br>four control (E12.5)<br>four mutant (E12.5) | Mean and SD | Ordinary one-way ANOVA | <0.0001 | F: 118.9<br>F(DFn, DFd): 0.3083 (3, 11) | Tukey's multiple comparative test | As indicated in the figure |
| 3-fs 3C | Three control (E11.5)<br>four mutant (E11.5)<br>three control (E12.5)<br>four mutant (E12.5) | Mean and SD | Ordinary one-way ANOVA | <0.0001 | F: 232.2<br>F(DFn, DFd): 0.4627 (3, 10) | Tukey's multiple comparative test | As indicated in the figure |
| 3-fs 3D | Four control (E11.5)<br>four mutant (E11.5)<br>four control (E12.5)<br>four mutant (E12.5) | Mean and SD | Ordinary one-way ANOVA | <0.0001 | F: 156.3<br>F(DFn, DFd): 0.9549 (3, 12) | Tukey's multiple comparative test | As indicated in the figure |
| 3-fs 3E | Four control (E11.5)<br>four mutant (E11.5)<br>four control (E12.5)<br>four mutant (E12.5) | Mean and SD | Ordinary one-way ANOVA | 0.1739 | F: 1.960<br>F(DFn, DFd): 0.4100 (3, 12) | Tukey's multiple comparative test | As indicated in the figure |
| 4-fs 2B | Four control (E13.5)<br>four mutant (E13.5)<br>four control (E14.5)<br>four mutant (E14.5)<br>four control (P0)<br>four mutant (P0) | Mean and SD | Ordinary one-way ANOVA | <0.0001 | F: 850.2<br>F(DFn, DFd): 3.630 (5, 18) | Tukey's multiple comparative test | As indicated in the figure |

*Table 2 continued on next page*

*Table 2 continued*

| Fig. | N | Descriptive statistics | Test used | p Value | Degrees of freedom and F/t/z/R/ETC | Pos hoc analysis | Adjusted p value |
|------|---|------------------------|-----------|---------|-------------------------------------|------------------|------------------|
| 5-fs 1B | Three control-OE mice 3 Pax2-OE mice | Mean and SD | Unpaired t-test (two-tailed) | <0.0001 | t = 22.79; df = 4 | - | - |
| 6-fs 1A | Three control mice four mutant mice | Mean and SD | Unpaired t-test (two-tailed) | <0.0001 | t = 12.82; df = 5 | - | - |
| 6-fs 1B | Three control mice four mutant mice | Mean and SD | Unpaired t-test (two-tailed) | 0.0002 | t = 10.1; df = 5 | - | - |

OE, overexpression; fs, figure supplement.

remove Reagent-1 and then placed into Reagent-2 (25% urea, 50% sucrose, 10% triethanolamine, 15% dH$_2$O) at 37°C for refractive index matching (3 days). Once the brains were cleared, they were imaged using a Zeiss Lightsheet Z.1 microscope. 3D reconstruction, photos and videos were created with arivis Vision4D.

### In utero electroporation

In utero electroporation was performed as previously described (*Saito and Nakatsuji, 2001*). Briefly, DNA plasmids were mixed with Fast Green and injected into the fourth ventricle of embryonic brains from outside the uterus with a glass micropipette. Holding the embryo in utero with forceps-type electrodes (NEPA GENE), 50 ms of 40 V electronic pulses were delivered five times at intervals of 950 ms with a square electroporator (Nepa Gene, CUY21). The plasmids used in this study are displayed in the Key Resource table. The primer sequences to clone to clone the mouse *Olig3* (NM_053008.3) and *Pax2* (NM_011037.5) genes are displayed in in the Key Resource table. The electroporated plasmid DNA mixtures were as follows: (i) for the control experiment, pCAG-GFP (0.5 mg ml$^{-1}$) + pCAG-Empty-IRES-GFP (0.5 mg ml$^{-1}$); (ii) for the *Olig3* overexpression experiment, pCAG-Olig3-IRES-GFP (0.5 mg ml$^{-1}$); and (iii) for the *Pax2* overexpression experiment, pCAG-Pax2-IRES-GFP (0.5 mg ml$^{-1}$).

### Statistics

Statistical analyses were performed using Prism 8 (GraphPad). Data are plotted in scatter dot plots or column dot plots with means and standard deviations (SD) displayed. The statistical significance between group means was tested by one-way ANOVA, followed by Tukey's post hoc test (for multiple- comparison tests), or two-tailed t-test (for pair comparison tests). Degrees of Freedom as well as F and t values are provided in *Table 2*. No statistical method was used to pre-determine the sample size. No randomization or blinding was used for *in vivo* studies.

## Acknowledgements

We thank Maarten Rikken and Fritz Rathjen for a critical reading of our manuscript. We are also grateful with Petra Stallerow and Claudia Päseler (at the Max-Delbrück-Center) as well as Koray Güner and Svetlana Tutukova (at the Charite Universitätsmedizin) for technical assistance. We also thank Prof. Carmen Birchmeier for providing us with *Olig3*$^{creERT2}$ and *Olig3*$^{GFP}$ mouse strains.

## Additional information

### Funding

| Funder | Grant reference number | Author |
|--------|------------------------|--------|
| Russian Science Foundation | 19-14-00345 | Victor Tarabykin |
| Fritz Thyssen Stiftung | 10.20.1.004MN | Luis R Hernandez-Miranda |

The funders had no role in study design, data collection and interpretation, or the decision to submit the work for publication.

## Author contributions

Elijah D Lowenstein, Conceptualization, Formal analysis, Investigation, Visualization, Methodology, Writing - review and editing; Aleksandra Rusanova, Jonas Stelzer, Investigation, Visualization, Methodology; Marc Hernaiz-Llorens, Adrian E Schroer, Francesca Bladt, Sven Buchert, Shiqi Jia, Investigation; Ekaterina Epifanova, Methodology; Eser Göksu Isik, Investigation, Visualization; Victor Tarabykin, Methodology, Visualization; Luis R Hernandez-Miranda, Conceptualization, Formal analysis, Supervision, Funding acquisition, Investigation, Visualization, Methodology, Writing - original draft, Project administration, Writing - review and editing

## Author ORCIDs

Elijah D Lowenstein  https://orcid.org/0000-0003-3755-0818
Marc Hernaiz-Llorens  http://orcid.org/0000-0003-1052-9613
Luis R Hernandez-Miranda  https://orcid.org/0000-0002-0498-708X

## Ethics

Animal experimentation: Animal experiments were approved by the local ethics committee LaGeSo (Landesamt für Gesundheit und Soziales) Berlin under animal experiment licenses G0026/14, and in accordance to the guidance and policies of the Charite Universitatsmedizin, Berlin, Germany; Max-Delbrück-Center for Molecular Medicine, Berlin, Germany; and the Institute of Neuroscience, Lobachevsky University of Nizhny Novgorod, Russian Federation.

## Decision letter and Author response

Decision letter https://doi.org/10.7554/eLife.64684.sa1
Author response https://doi.org/10.7554/eLife.64684.sa2

# Additional files

## Supplementary files
• Transparent reporting form

## Data availability

All data generated or analysed during this study are included in the manuscript and supporting files.

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
