## [Decision Letter]

**Acceptance summary:**

This paper presents exciting evidence that the bHLH transcription factor Olig3 controls the development of both cerebellar germinal zones. The data therefore have major implications for how GABAergic and glutamatergic neurons are produced.

**Decision letter after peer review:**

Thank you for submitting your article "Olig3 acts as a master regulator of cerebellar development" for consideration by *eLife*. Your article has been reviewed by three peer reviewers, and the evaluation has been overseen by a Reviewing Editor and Marianne Bronner as the Senior Editor. The following individuals involved in review of your submission have agreed to reveal their identity: Victor V Chizhikov (Reviewer #1); David J Solecki (Reviewer #2); Kimberly A Aldinger (Reviewer #3).

The reviewers have discussed the reviews with one another and the Reviewing Editor has drafted this decision to help you prepare a revised submission.

Summary:

This work identifies the bHLH transcription factor Olig3 as a protein expressed in the germinal zones of the developing mouse cerebellum. The authors used a combination of histology as well as quantification of immunohistochemistry data, fate mapping, and in utero electroporation in control and mutant mice to test the role of Olig3 during cerebellar development. The data demonstrate key functions for Olig3+ in cells of the rhombic lip and ventricular zone, the two progenitor zones that give rise to cerebellar glutamatergic and GABAergic neurons, respectively.

Essential revisions:

The reviewers have made suggestions on how to improve the clarity of specific figures. Please see below for details. In addition, there are a number of suggestions to improve the clarity of how the data are interpreted. Therefore, there are minor additional data analysis/experiments required and several instances of reworking the text.

Title:

Please consider revising the title. The use of "master regulator" in particular could be toned down. Perhaps "early" regulator may be more appropriate.

Reviewer #1:

Lowenstein et al., describe the novel roles of Olig3 in cerebellar development. The authors showed that Olig3 is predominantly required for the proper development of the earliest ventricular zone (VZ) and rhombic lip (RL) derivatives. They found that Olig3 regulates proliferation in the RL. Also, they showed that Olig3 acts cell-autonomously to prevent the misspecification of VZ-derived Purkinje cells (PCs) into inhibitory interneurons, by suppressing the expression of Pax2. Currently, the mechanisms that segregate different VZ and RL-derived lineages are poorly understood, and the implication of Olig3 in the segregation of PCs and inhibitory interneurons (with underlying mechanisms) is the most exciting part of this paper. As a whole, the paper makes a very valuable addition to our understanding of the mechanisms of cerebellar development.

In general, the experimental analysis is well done, and most conclusions are supported by strong experimental data. However, there are several cases of the over-interpretation of findings, which need to be addressed by additional analysis, more accurate discussion, and/or by providing better images.

1) Please provide higher magnification images (insets) to better appreciate the presence of cells double-positive for TdTomato/Brn2 and TdTomato/Tbr1 (Figure 2D); and cells double-positive for b-gal/Pax6 (Figure 2E), b-gal/Foxp2 (Figure 2G), and b-gal/Tbr1 (Figure 2F), and the lack of b-gal and Pax2 overlap (Figure 2H).

2) Figure 3—figure supplement 3C: based on the provided panels, it is hard to see a difference in the number of BrdU^+^ cells in the RL of *Olig3^GFP/GFP^* and control embryos (region 2 in the mutant versus region 2 in the control embryo). More representative images need to be shown.

3) Throughout the paper, the authors state that loss of Olig3 results in ectopic expression of Pax2 in the RL (for example in the Introduction and Results). However, both Figure 4C (boxed regions) and Figure 4—figure supplement 1A (region 2) show area adjacent to the RL, not the RL itself. To conclude that loss of Olig3 induces ectopic Pax2 expression in the RL, the authors need to show a high magnification of the RL itself. To precisely determine RL limits, Pax2 staining needs to be combined with a RL marker, such as Atoh1 (Pax2/Atoh1 double labeling or at least Pax2 and Atoh1 labeling in adjacent sections).

Alternatively, the authors may remove their conclusion regarding Pax2 de-repression in the RL of Olig3 mutants and delete a related discussion of inhibitory interneurons as the default neuronal phenotype (Discussion).

4) Figure 4C: It is hard to see whether in Olig3 mutant (the lower right high magnification image) Pax2+ cells co-express Foxp2. Single channel images are needed in addition to the already presented overlayed (red+blue channels together) image.

5) (Results and Discussion). The authors conclude that "Pax2.… imposes an inhibitory interneuron fate". However, in this paper it is only shown that Pax2 inhibits Foxp2 (PC fate). To conclude that Pax2 is sufficient to impose an inhibitory interneuron fate, additional markers of mature interneurons need to be analyzed in Pax2-electroporated cells (in embryos in utero electroporated with a Pax2 expressing vector). Alternatively, authors can delete the statement about Pax2 imposing interneuronal fate and simply state that Pax2 is sufficient to suppress PC (Foxp2) identity.

6) It needs to be discussed that granule cell defects may be secondary to PC defects in Olig3 mutants rather than result from a direct role of Olig3 in the development of granule cell lineage.

Reviewer #2:

I don't have any further suggestions for this well-done study.

Reviewer #3:

In this study, the authors identify Olig3 among bHLH transcription factors expressed in the rhombic lip of the developing cerebellum using data from the Allen Developing Mouse Brain Atlas. They then use histology and IHC with quantitation, fate mapping, and in utero electroporation in wildtype and mutant mice to evaluate the role of Olig3 in early cerebellar development. They show Olig3+ cells in RL and VZ, the two progenitor zones that give rise to glutamatergic and GABAergic cells, respectively. Lineage tracing using *Olig3^creERT2^* with Ai14 or TaunLacZ confirmed Olig3+ progenitors give rise to early cells that arise from the RL and VZ. Olig3 KO mice had a small, minimally foliated cerebellar vermis at birth. Most cell types were reduced in number consistent with overall hypoplasia, but Pax2+ cells were increased in number. The authors confirmed Olig3 absence leads to a fate change in cells emanating from the VZ with forced overexpression of Olig3 leading to Olig3+/Pax2+ cells. The further confirm that Olig2 does not compensate for this phenotype.

This study newly identifies Olig3 as an important determinant of early cerebellar cell types emanating from the RL and VZ. This paper is well written, methods are sound, and the Results and Discussion present the relationship to prior studies.

1) How was the list of bHLH transcription factors compiled? Does 108 represent ALL bHLH or only the subset that have data available in the Allen atlas? Are there additional bHLH that are not available in Allen? Please provide additional details. E.g. To orient the reader, it would be helpful to provide examples of the Allen gene expression for one gene that represents each category listed in Table 1/Figure 1A.

2) How are RL and VZ defined in this study? There are no anatomic boundaries that delineate these transient, three dimensional structures; how were boundaries determined? Please provide details for how these regions were identified (e.g. Atoh1+, PMID: 16202705) and quantified.

3) The Olig3+/Sox2Sox2^+^ double positive yellow cells are difficult to see in Figure 1C. Showing an image as in Figure 1E could help the reader to visually correlate the quantitation of the double positive cells.

4) In the Discussion the authors state that Olig3 loss reduces the number of Atoh1+ cells. However, no expression of Atoh1 in Olig3 KO is shown to support this conclusion. The majority of Olig3+ RL cells do not express Atoh1 (Figure 1D). It is not clear from the data shown that Olig3 KO specifically affects Atoh1+ cells in the RL. Lmx1a is another gene critical for RL progenitor generation that is not co-expressed with Atoh1 (e.g. PMID: 204098) and Lmx1a/Olig3 coexpression was not evaluated; it's plausible that the other 2/3 of the Olig3+ RL cells coexpress Lmx1a.

---

## [Author Response]

Essential revisions:The reviewers have made suggestions on how to improve the clarity of specific figures. Please see below for details. In addition, there are a number of suggestions to improve the clarity of how the data are interpreted. Therefore, there are minor additional data analysis/experiments required and several instances of reworking the text.Title:Please consider revising the title. The use of "master regulator" in particular could be toned down. Perhaps "early" regulator may be more appropriate.

We have considered your suggestion regarding revising the title and replaced the world “master” with “early”. The title of our work now reads:

“Olig3 regulates early cerebellar development”.

Reviewer #1:Lowenstein et al., describe the novel roles of Olig3 in cerebellar development. The authors showed that Olig3 is predominantly required for the proper development of the earliest ventricular zone (VZ) and rhombic lip (RL) derivatives. They found that Olig3 regulates proliferation in the RL. Also, they showed that Olig3 acts cell-autonomously to prevent the misspecification of VZ-derived Purkinje cells (PCs) into inhibitory interneurons, by suppressing the expression of Pax2. Currently, the mechanisms that segregate different VZ and RL-derived lineages are poorly understood, and the implication of Olig3 in the segregation of PCs and inhibitory interneurons (with underlying mechanisms) is the most exciting part of this paper. As a whole, the paper makes a very valuable addition to our understanding of the mechanisms of cerebellar development.In general, the experimental analysis is well done, and most conclusions are supported by strong experimental data. However, there are several cases of the over-interpretation of findings, which need to be addressed by additional analysis, more accurate discussion, and/or by providing better images.1) Please provide higher magnification images (insets) to better appreciate the presence of cells double-positive for TdTomato/Brn2 and TdTomato/Tbr1 (Figure 2D); and cells double-positive for b-gal/Pax6 (Figure 2E), b-gal/Foxp2 (Figure 2G), and b-gal/Tbr1 (Figure 2F), and the lack of b-gal and Pax2 overlap (Figure 2H).

We now provide magnifications for the suggested figures.

In addition, we provide similar magnifications in Figure 2—figure supplement 1D and E.

2) Figure 3—figure supplement 3C: based on the provided panels, it is hard to see a difference in the number of BrdU^+^ cells in the RL of Olig3^GFP/GFP^ and control embryos (region 2 in the mutant versus region 2 in the control embryo). More representative images need to be shown.

In the revised Figure 3—figure supplement 3C we now provide more representative images with insets displaying BrdU signals.

3) Throughout the paper, the authors state that loss of Olig3 results in ectopic expression of Pax2 in the RL (for example in the Introduction and Results). However, both Figure 4C (boxed regions) and Figure 4—figure supplement 1A (region 2) show area adjacent to the RL, not the RL itself. To conclude that loss of Olig3 induces ectopic Pax2 expression in the RL, the authors need to show a high magnification of the RL itself. To precisely determine RL limits, Pax2 staining needs to be combined with a RL marker, such as Atoh1 (Pax2/Atoh1 double labeling or at least Pax2 and Atoh1 labeling in adjacent sections).Alternatively, the authors may remove their conclusion regarding Pax2 de-repression in the RL of Olig3 mutants and delete a related discussion of inhibitory interneurons as the default neuronal phenotype (Discussion).

Dr. Victor V. Chizhikov is right in that our data did not directly address the fact that Pax2 becomes de-repressed in the rhombic lip. Methodologically, we face the problem that our Atoh1 and Pax2 antibodies are both generated in rabbit, which prevented us from co-staining Olig3 mutant tissue with both antibodies to determine whether Atoh1+ cells (like Foxp2+ cells) might co-express Pax2 in Olig3 mutant mice. Consecutive sections separately stained with Pax2 and Atoh1 do show that Pax2+ cells locate inside the Atoh1 domain in Olig3 mutant mice. However, we now reason that this result alone cannot unambiguously define whether these cells indeed generate from the rhombic lip or whether these cells are generated from the ventricular zone and displaced into the rhombic lip. We therefore removed our conclusion that Pax2 becomes de-repressed in the rhombic lip of Olig3 mutant mice throughout the text.

4) Figure 4C: It is hard to see whether in Olig3 mutant (the lower right high magnification image) Pax2+ cells co-express Foxp2. Single channel images are needed in addition to the already presented overlayed (red+blue channels together) image.

We now provide a new figure, Figure 4—figure supplement 1, illustrating additional examples of Pax2 and Foxp2 co-expression in control and Olig3 mutant mice, with magnifications and the display of single and merged fluorescent channels. We did not alter the original Figure 4C as the main point we want to show in this figure is the de-repression of Pax2 in GFP+ cells in *Olig3^GFP/GFP^* mutant mice.

5) (Results and Discussion). The authors conclude that "Pax2.… imposes an inhibitory interneuron fate". However, in this paper it is only shown that Pax2 inhibits Foxp2 (PC fate). To conclude that Pax2 is sufficient to impose an inhibitory interneuron fate, additional markers of mature interneurons need to be analyzed in Pax2-electroporated cells (in embryos in utero electroporated with a Pax2 expressing vector). Alternatively, authors can delete the statement about Pax2 imposing interneuronal fate and simply state that Pax2 is sufficient to suppress PC (Foxp2) identity.

Dr. Victor V. Chizhikov is right that our original data only showed the suppressive function of Pax2 on Foxp2 expression.

We stained the cerebella of wildtype mice at E14.5 and E15.5 with antibodies against classical markers of GABAergic cells such as Parvalbumin, Somatostatin, Calbindin, Claretinin, Gad67 and Gad65, and compared their expression against Foxp2 (that labels Purkinje cells). We found Parvalbumin as a candidate marker for newborn inhibitory interneurons as it i) does not colocalize with Foxp2, and ii) follows that same distribution and expression pattern as Pax2, at least during the analyzed ages. We co-stained control and Pax2-electroporated cells with Parvalbumin and found that about a third of the Pax2-electroporated cells co-express Parvalbumin, whereas control electroporated cells rarely do. We now add this data in Figure 5—figure supplement 1 and in the Results section.

However, we modified our conclusion that Pax2 imposes an inhibitory interneuron fate, as we now reason that the most conclusive way to demonstrate this would be by analyzing the morphology and physiology of Pax2-electroporated cells in postnatal life, which is outside the scope of our present work.

We now conclude:

“We conclude that Pax2 is an efficient suppressor of Foxp2 expression and that its expression seems to induce a differentiation program characteristic of inhibitory interneurons.”

We also modified the heading of this this experimental Results section to say:

“Olig3 cell-autonomously curtails Pax2 expression to secure Purkinje cell differentiation”.

6) It needs to be discussed that granule cell defects may be secondary to PC defects in Olig3 mutants rather than result from a direct role of Olig3 in the development of granule cell lineage.

We thank the reviewer for this insight. We now discuss this as follows:

“One should note, however, that the reduced numbers of granule cells in Olig3 mutant mice might be mainly independent of Olig3 and due to the reduction of instructive signals emanating from Purkinje cells, which are severely affected in these mutant mice. Indeed, available evidence shows that Purkinje cells regulate EGL proliferation and the differentiation of granule cells via sonic hedgehog signaling (Dahmane and Ruiz i Altaba, 1999; Wallace, 1999; Wechsler-Reya and Scott, 1999).”

Reviewer #3:In this study, the authors identify Olig3 among bHLH transcription factors expressed in the rhombic lip of the developing cerebellum using data from the Allen Developing Mouse Brain Atlas. They then use histology and IHC with quantitation, fate mapping, and in utero electroporation in wildtype and mutant mice to evaluate the role of Olig3 in early cerebellar development. They show Olig3+ cells in RL and VZ, the two progenitor zones that give rise to glutamatergic and GABAergic cells, respectively. Lineage tracing using Olig3^creERT2^ with Ai14 or TaunLacZ confirmed Olig3+ progenitors give rise to early cells that arise from the RL and VZ. Olig3 KO mice had a small, minimally foliated cerebellar vermis at birth. Most cell types were reduced in number consistent with overall hypoplasia, but Pax2+ cells were increased in number. The authors confirmed Olig3 absence leads to a fate change in cells emanating from the VZ with forced overexpression of Olig3 leading to Olig3+/Pax2+ cells. The further confirm that Olig2 does not compensate for this phenotype.This study newly identifies Olig3 as an important determinant of early cerebellar cell types emanating from the RL and VZ. This paper is well written, methods are sound, and the Results and Discussion present the relationship to prior studies.1) How was the list of bHLH transcription factors compiled? Does 108 represent ALL bHLH or only the subset that have data available in the Allen atlas? Are there additional bHLH that are not available in Allen? Please provide additional details. E.g. To orient the reader, it would be helpful to provide examples of the Allen gene expression for one gene that represents each category listed in Table 1/Figure 1A.

Following the suggestion above we created a new figure, Figure 1—figure supplement 1, to include examples from the Allen Developing Mouse Brain Atlas of two genes that represent each category listed in Table 1/Figure 1A.

The precise number of bHLH transcription factors expressed in humans as well as in other organisms is yet to be defined. Phylogenetic analyses suggest that humans express approximately 130 bHLH factors, whereas mice express about 117 (Skinner et al., 2010; Stevens et al., 2008). In this study, we used the 110 genes annotated in the Human Genome Organization (HuGO; https://www.genenames.org/data/genegroup/#!/group/420). In our original search on the Allen Developing Mouse Brain Atlas we found data for 108/110 of the HuGO annotated genes. The genes that were not found in our original search were Arntl and Mycl.

During the revision of our manuscript, we revisited the Allen Brain Atlas and found data for both genes. We now include them in our list and classify them as not expressed (Arntl) and generically expressed in cerebellar progenitor cells (Mycl). The following changes have been made in our revised manuscript:

i) In the Results, we describe the selection of the 110 bHLH factors that are annotated in HuGO and provide the link for the resource.

ii) In the Results, we adjust our original number of listed genes to include Arntl and Mycl1 according to their expression.

iii) Figure 1A has been modified from 108 genes to 110 genes.

iv) Table 1 now includes Arntl1 and Mycl.

2) How are RL and VZ defined in this study? There are no anatomic boundaries that delineate these transient, three dimensional structures; how were boundaries determined? Please provide details for how these regions were identified (e.g. Atoh1+, PMID: 16202705) and quantified.

We defined RL and VZ boundaries according to the expression of Atoh1 and Ptf1a, respectively. To help the reader to visualize these boundaries, we now provide a revised Figure 1—figure supplement 2, in which panel B illustrates the expression of Olig3 together with Atoh1 and Ptf1a.

3) The Olig3+/Sox2Sox2^+^ double positive yellow cells are difficult to see in Figure 1C. Showing an image as in Figure 1E could help the reader to visually correlate the quantitation of the double positive cells.

In our revised Figure 1C, we now show magnifications of the rhombic lip and ventricular zone. Furthermore, we now display the images of the ventricular zone similar to the image of Figure 1E. We also displayed the Sox2 signals in blue to improve the visualization of cells co-expressing Olig3 (displayed in red). We did not include insets for the rhombic lip, as we believe the colocalization is easy to distinguish in its current form.

4) In the Discussion the authors state that Olig3 loss reduces the number of Atoh1+ cells. However, no expression of Atoh1 in Olig3 KO is shown to support this conclusion. The majority of Olig3+ RL cells do not express Atoh1 (Figure 1D). It is not clear from the data shown that Olig3 KO specifically affects Atoh1+ cells in the RL. Lmx1a is another gene critical for RL progenitor generation that is not co-expressed with Atoh1 (e.g. PMID: 204098) and Lmx1a/Olig3 coexpression was not evaluated; it's plausible that the other 2/3 of the Olig3+ RL cells coexpress Lmx1a.

In our original Figure 3—figure supplement 3A, we showed immunostaining and quantification of Atoh1+ cells in Olig3 KO mice at E11.5 and at E12.5. These data illustrated a reduction in the number of Atoh1+ cells in Olig3 KO mice. We now also provide insets in the original photographs without GFP to improve the visualization of Atoh1+ cells.

In this study we focused on the elucidation of bHLH transcription factors responsible for the generation of the distinct cerebellar neuron types and did not include homeodomain proteins such as Lmx1a. Following the observation of reviewer 3, we performed fluorescent in situ hybridization for Olig3 and Lmx1a transcripts. This shows that indeed both Olig3 and Lmx1a partially co-localize in the rhombic lip (see Author response image 1). The potential cooperativity and interaction between Olig3 and Lmx1a in development of early rhombic lip derivatives is out of the scope of this work and remains to be determined.

**Author response image 1. sa2fig1:** Fluorescent in situ hybridization for Olig3 and Lmx1a transcripts.